# INFERENCE-TIME ALIGNMENT OF DIFFUSION MODELS WITH DIRECT NOISE OPTIMIZATION

## ABSTRACT

In this work, we focus on the alignment problem of diffusion models with a continuous reward function, which represents specific objectives for downstream tasks, such as increasing darkness or improving the aesthetics of images. The central goal of the alignment problem is to adjust the distribution learned by diffusion models such that the generated samples maximize the target reward function. We propose a novel alignment approach, named Direct Noise Optimization (DNO), that optimizes the injected noise during the sampling process of diffusion models. By design, DNO operates at *inference-time*, and thus is *tuning-free* and *prompt-agnostic*, with the alignment occurring in an online fashion during generation. We rigorously study the theoretical properties of DNO and also propose variants to deal with non-differentiable reward functions. Furthermore, we identify that naive implementation of DNO occasionally suffers from the *out-of-distribution reward hacking* problem, where optimized samples have high rewards but are no longer in the support of the pretrained distribution. To remedy this issue, we leverage classical high-dimensional statistics theory to an effective probability regularization technique. We conduct extensive experiments on several important reward functions and demonstrate that the proposed DNO approach can achieve state-of-the-art reward scores within a reasonable time budget for generation.

## 1 INTRODUCTION

Diffusion models work by learning to reverse the process of diffusing the data distribution $p(x)$ into noise, which can be described by a stochastic differential equation (SDE) (Song et al., 2020b; Karras et al., 2022): $dx_t = f(t)x_t dt + g(t)dw_t$, where $dw_t$ is the standard Wiener process, and $f(t)$ and $g(t)$ are the drift and diffusion coefficients, respectively. The reverse process relies on the score function $\epsilon(x_t, t) \stackrel{\text{def.}}{=} \nabla_x \log p_t(x)$ where $p_t$ denotes the p.d.f of noisy data $x_t$, and its closed-form can be expressed either as an ODE or as an SDE: (Song et al., 2020b):

$$\text{ODE:} \quad dx_t = \left( f(t)x_t - \frac{1}{2}g^2(t)\epsilon(x_t, t) \right) dt, \tag{1}$$

$$\text{SDE:} \quad dx_t = \left( f(t)x_t - g^2(t)\epsilon(x_t, t) \right) dt + g(t)dw_t. \tag{2}$$

With the capability to evaluate $\epsilon(x_t, t)$, it becomes possible to generate samples from noise by numerically solving either the ODE equation 1 or the SDE equation 2. The training process, therefore, involves learning a parameterized surrogate $\epsilon_\theta(x_t, t)$ to approximate $\epsilon(x_t, t)$, following a denoising score matching framework as described in (Song et al., 2020b; Karras et al., 2022). Despite the effectiveness of diffusion models in modeling continuous distributions, when deploying these generative models for specific tasks, it is not suitable to sample from the original learned distribution directly, because this distribution has *not been aligned* with the task-specific objective. For instance, in image generation, users may wish to produce images that are aesthetically pleasing rather than mediocre, or generate images with enhanced brightness, darkness, or compressibility. Recently, the alignment problem has drawn considerable interest in the context of diffusion models, as evidenced by a series of studies such as (Yuan et al., 2024; Song et al., 2023a; Dong et al., 2023; Prabhudesai et al., 2023; Black et al., 2023; Fan et al., 2023).

**Alignment Problem for Diffusion Models.** Given a diffusion model characterized by parameters $\theta$ and its associated distribution $p_\theta(x)$, as well as a reward function $r(x)$ that can assign real-valued scores to generated samples, the central goal of the alignment problem is to adjust the distribution $p_\theta(x)$ such that the generated samples maximize the reward from $r(x)$. In this work, we consider the reward functions to be continuous but possibly non-differentiable. In the following sections, we will provide a comprehensive review of some well-established methods for aligning diffusion models.

## 1.1 METHODS FOR ALIGNING DIFFUSION MODELS

### 1.1.1 ONLINE/OFFLINE REINFORCEMENT LEARNING FOR FINE-TUNING DIFFUSION MODELS

A common mathematical formulation in RL-based methods is to maximize the expected reward while ensuring the resulting distribution does not deviate excessively from the original distribution. This can be expressed as the following KL-regularized optimization problem: $\max_p \mathbb{E}_{x \sim p(x)}[r(x)] - \lambda \text{KL}(p||p_\theta)$. Current RL-based methods can be categorized into online and offline methods based on the data used. In the online method, the algorithm has the capacity to query the reward function throughout the entire optimization process. Two notable online RL methods are DDPO (Black et al., 2023) and DPOK (Fan et al., 2023), which have been shown to improve downstream objectives such as aesthetics and compressibility. Alternatively, research has also delved into the offline RL optimization setting, where an explicit reward function is not accessible and only a fixed preference dataset is utilized. Noteworthy works in this category include Diffusion-DPO (Wallace et al., 2023a) and SPIN-Diffusion (Yuan et al., 2024).

### 1.1.2 DIRECT FINE-TUNING OF DIFFUSION MODELS WITH DIFFERENTIABLE REWARDS

Before delving into the formal description of this method, it is useful to revisit the sampling process of diffusion models, which also serves as the foundation for the rest of this work. Solving the ODE equation 1 or the SDE equation 2 typically involves discretizing the time steps into $T$ steps. By starting with the initial noise $x_T \sim \mathcal{N}(0, I)$, the solution process for the ODE equation 1 can be viewed as a mapping that transforms the initial noise $x_T$ into less noisy data through the following process: $x_{t-1} = \text{ODE\_solver}(x_t)$, for $t = T, ..., 1$. After $T$ steps, the output will be the generated sample $x_0$. Similarly, solving the SDE equation 2 can be seen as a mapping that gradually converts *both* the initial noise $x_T$ and the entire extra random noise vectors $z_T, ..., z_1$ into the generated sample $x_0$, e.g., through the following process: $x_{t-1} = \text{SDE\_solver}(x_t, z_t)$, for $t = T, ..., 1$, where $z_t$ is also drawn from standard Gaussian distribution.

**Remark 1.** *Throughout this work, for simplicity, we adopt only the DDIM sampling algorithm (Song et al., 2020a) for our experiments, as it remains one of the most popular choices for diffusion sampling and, more importantly, supports both ODE-style and SDE-style sampling. To be self-contained, we summarize the notations and procedure of the DDIM sampling method in Appendix A.*

**Diffusion Sampling as a Noise-to-Sample Mapping.** From the diffusion sampling process described above, we observe that the sampling process can be conceptualized as an end-to-end mapping $M_\theta(z)$, which translates noise vectors $z$, sampled from the standard Gaussian distribution, into generated samples. Here, the noise vectors $z$ serve as a unified abstraction for both the $x_T$ in the ODE-based sampling process and the $(x_T, \{z_1, ..., z_T\})$ in the SDE-based sampling process. As we can see, the noise vector $z$ uniquely determines the generated sample from the diffusion models.

Two recent studies, AlignProp (Prabhudesai et al., 2023) and DRaFT (Clark et al., 2023), have proposed to directly fine-tune diffusion models using differentiable rewards. Specifically, their optimization objective is formulated as: $\max_\theta \mathbb{E}_{z \sim \mathcal{N}(0,I)}[r(M_\theta(z))]$. Both the AlignProp (Prabhudesai et al., 2023) and DRaFT (Clark et al., 2023) methods utilize the ODE-type DDIM solver for the sampling process, specifically employing Algorithm 1 with $\eta = 0$.

### 1.1.3 LOSS-GUIDED DIFFUSION

A recent work focusing on loss-guided diffusion models (LGD) (Song et al., 2023a) also examines the concept of aligning diffusion models with differentiable rewards. Unlike the methods mentioned previously, LGD is an **Inference-Time** alignment method, meaning it does not necessitate modifications to the pretrained model $\theta$ and only works by modifying the inference process. In essence, the core idea of LGD is that, during the sampling process for the ODE equation 1, it considers a modified version of the ODE by introducing a new term that guides the generation toward areas of higher reward. Specifically, the new ODE is: $dx_t = \left(f(t)x_t - \frac{1}{2}g^2(t)\epsilon(x_t, t) + \nabla_{x_t} r(x_0(x_t))\right) dt$, where $x_0(x_t)$ denotes the solution of this ODE starting from $x_t$. However, the gradient term $\nabla_{x_t} r(x_0(x_t))$ is not readily available during generation. To address this, the authors suggest utilizing Monte Carlo estimation to approximate the gradient. Nevertheless, this estimation tends to be noisy and imprecise, leading to suboptimal performance, particularly with complex reward functions.

## 1.2 COMPARING EXISTING METHODS

Existing works can be generally categorized based on two criteria: whether it requires fine-tuning and whether the reward function needs to be differentiable.

**Inference-Time or Tuning-based Methods.** All RL-based methods and the direct fine-tuning method are tuning-based, meaning they necessitate adjustments to the network models $\theta$. There

are two main disadvantages associated with tuning-based methods. The first is that they require fine-tuning for new reward functions, a process that can consume considerable resources, especially when faced with extensive choices for reward functions. The second drawback is that the fine-tuning process typically relies on a limited set of input prompts, which challenges the model to generalize to new and unseen prompts. In contrast, methods such as LGD (Song et al., 2023a) belong to the inference-time category. The main advantage of the inference-time approach is its elimination of the need for fine-tuning, as well as its ability to avoid the prompt generalization issues associated with tuning-based methods. This makes it **prompt-agnostic**, as the inference-time method optimizes the sample specifically for the given prompt during the inference process. Further, inference-time methods require significantly fewer computing resources than tuning-based methods. However, the major drawback of inference-time methods is the substantial increase in the time required for the generation process.

**Differentiable or Non-Differentiable Rewards.** Current RL-based methods can work by utilizing solely the value or preference information of the reward functions, therefore they can still work even when the reward function is non-differentiable. In contrast, the existing direct fine-tuning methods and LGD require the reward function to be differentiable. In practice, working with non-differentiable reward functions is important due to their prevalence. This non-differentiable property can arise from the simulation-based procedures used to compute the reward, or the reward function itself may be a black box.

For additional discussion on existing literature, please refer to Appendix G.

### 1.3 OUR CONTRIBUTIONS

In this work, we focus on inference-time alignment of diffusion models, as we believe that flexibility with the choices of the reward functions, generalization on unseen prompts, and low computing requirements are critical for a broad range of real-world applications. Our primary goal is to design an inference-time alignment method that can match the performance of tuning-based methods by incurring a reasonable additional time cost, and is capable of handling both differentiable and non-differentiable objective functions. To this end, we focus on an under-explored technique for achieving inference-time alignment of diffusion models—**Direct Noise Optimization (DNO)**. Specifically, we make the following contributions:

- We conduct a self-contained and comprehensive study for DNO, and demonstrate that noise optimization can be theoretically interpreted as sampling from an improved distribution.
- We identify out-of-distribution reward-hacking as a critical issue in DNO. To address this issue, we introduce a novel probability-regularized noise optimization method designed to ensure the generated samples remain within the support of pretrained distribution.
- By developing a novel and highly efficient hybrid gradient approximation strategy, we extend the DNO approach to handle non-differentiable reward functions effectively.
- Through the experiments on several important image reward functions, we demonstrate that our proposed method can achieve state-of-the-art scores in comparison to existing alignment methods, without any fine-tuning on the parameters of diffusion models.

## 2 DIRECT NOISE OPTIMIZATION FOR ALIGNING DIFFUSION MODELS

Given the noise-to-sample mapping $M_\theta$ described in Section 1.1.2, DNO can be mathematically formulated as follows:

$$\max_z \ r(M_\theta(z)), \tag{3}$$

with $z \sim \mathcal{N}(0, I)$ as the initial solution. As we will discuss in Section 3, the Gaussian distribution $\mathcal{N}(0, I)$ serves as an important prior on the optimization variables $z$. By solving this optimization problem, we can obtain the optimized noise vectors, which are then used to generate high-reward samples. When the reward function $r(\cdot)$ is differentiable, gradient-based optimization methods can be applied to solve the problem efficiently. That is, the following step can be performed iteratively until convergence: $z_{\text{new}} = \text{optimizer\_step}(z_{\text{old}}, \nabla_z r\left(M_\theta(z_{\text{old}})\right))$, where the optimizer can be either vanilla gradient ascent or adaptive optimization algorithms like Adam (Kingma & Ba, 2014). When the gradient of reward function $r(\cdot)$ is not available, we can leverage techniques from zeroth order optimization (Nesterov & Spokoiny, 2017; Tang et al., 2024a) to estimate the ground-truth gradient $\nabla_z r\left(M_\theta(z_{\text{old}})\right)$, denoted as $\hat{g}(z_{\text{old}})$, and then apply similarly $z_{\text{new}} = \text{optimizer\_step}(z_{\text{old}}, \hat{g}(z_{\text{old}}))$. In Section 4, we provide a dedicated discussion on how to obtain a better estimator for $\hat{g}(z_{\text{old}})$ when the reward function is non-differentiable.

Several recent studies have explored similar formulations to equation 3 across different applications. (Wallace et al., 2023b) investigates the optimization of latent vectors obtained through DDIM-inversion (Song et al., 2020a), aiming to enhance the CLIP score (Radford et al., 2021) and Aesthetic Score (Schuhmann et al., 2022b) of given images. (Ben-Hamu et al., 2024) discusses optimizing the initial noise $x_T$ for the ODE process to address inverse problems, leveraging the diffusion model as a prior. (Novack et al., 2024) and (Karunratanakul et al., 2023) consider the optimization of initial noise $x_T$ for the ODE with the objective of improving downstream objectives in robotics and audio.

While similar methods of DNO has appeared in previous works, many of its technical details remain insufficiently explored. On one hand, there is a lack of a comprehensive framework concerning the design choices, theoretical understanding, and practical challenges associated with DNO for aligning diffusion models. On the other hand, the field has yet to systematically investigate whether DNO, as a inference-time method for aligning diffusion models, can achieve competitive performance compared to tuning-based methods. In this work, we aim to conduct a thorough study on DNO. In the following two sections, we dive deep to understand the theoretical foundation of DNO and discuss a critical design choice ignored in previous works.

### 2.1 UNDERSTANDING DIRECT NOISE OPTIMIZATION

In Appendix B.1, we present a simple example to visualize the process of DNO, where we observed that the distribution of the generated samples shifts toward a distribution on the local maxima of the reward function. Inspired by this example, we propose to view **DNO as sampling from an improved distribution.** To rigorously describe this evolving process, we define an operator function $g$ to represent a single gradient step, i.e., $g(z) \stackrel{\text{def}}{=} z + \ell \cdot \nabla_z r \circ M_\theta(z)$, where $\circ$ denotes the function composition operator and $\ell$ denotes the step size for gradient ascent. Additionally, we define the operator $g_t$, which denotes applying the gradient ascent step for $t$ steps, i.e., $g_t(z) = g(g_{t-1}(z))$ with $g_0$ being the identity mapping. With these notations, we can now express the distribution after $t$ gradient steps as $p_t(x)$, which is the distribution of $M_\theta(g_t(z))$ with $z \sim \mathcal{N}(0, I)$. In the following theorem, we demonstrate that there is a rigorous improvement after every single gradient step, i.e., the distribution $p_{t+1}(x)$ is provably better than $p_t(x)$ in terms of expected reward.

**Theorem 1.** *Assuming that $r \circ M_\theta$ is $L$-smooth, namely, $\|\nabla r \circ M_\theta(z) - \nabla r \circ M_\theta(z')\| \leq L\|z - z'\|$ for any $z \neq z'$, it holds true that*

$$\mathbb{E}_{x \sim p_{t+1}(x)} r(x) \geq \mathbb{E}_{x \sim p_t(x)} r(x) + \left(\ell - \frac{\ell^2 L}{2}\right) \mathbb{E}_{z_0 \sim N(0,I)} \left\|\nabla_z r \circ M_\theta(z)_{|z = g_t(z_0)}\right\|_2^2. \quad (4)$$

In Theorem 1, we rely on the smoothness assumption of the composite mapping $r \circ M_\theta$. We note that this is a reasonable assumption in practice, as the noise-to-sample mapping in diffusion models has been observed to be smooth. For instance, see Figure 4 in (Tang et al., 2024a). We also provide a justification for the smoothness of reward function in Appendix H. As described in equation 4, provided that the step size $\ell$ is less than $\frac{2}{L}$, the distribution $p_{t+1}(x)$ is strictly better than the previous distribution $p_t(x)$ in terms of expected reward, as long as the second term is non-zero. Based on this result, it is natural to ask: *When does the distribution stop improving?* Namely, when does the second term in equation 4 become zero. We provide a detailed discussion to answer this question and also the proof for Theorem 1 in Appendix C.

### 2.2 OPTIMIZING ODE VS. OPTIMIZING SDE

As previously introduced, there are two primary methods for sampling from pretrained diffusion models: one based on solving the ODE equation 1 and the other on solving the SDE equation 2. A critical difference lies in the fact that ODE sampling depends exclusively on the initial noise $x_T$, whereas SDE sampling is additionally influenced by the noise $z_t$ added at every step of the generation process. It has been noted that existing works on noise optimization (Ben-Hamu et al., 2024; Novack et al., 2024; Karunratanakul et al., 2023) have mainly concentrated on optimizing the initial noise $x_T$ for the ODE sampler.

Different from preceding studies, we explore the utilization of the SDE sampler for noise optimization. Specifically, we employ the DDIM with $\eta = 1$ (Song et al., 2020a) as the SDE sampler and propose to optimize both the added noise $z_t$ at every timestep and the initial noise $x_T$. In this context, the dimensionality of the optimized noise significantly surpasses that of ODE sampler, typically $T \cdot d$ v.s. $d$, where $d$ is the dimension for the learned

Figure 1: ODE vs. SDE for optimization

distribution and $T$ is the number of discretization steps in the sampling process. With this higher dimension for optimization, we have empirically observed that optimizing the SDE-based generation process can be significantly better than its ODE-based counterpart. This is illustrated in Figure 1, where we juxtapose the optimization speeds between the ODE (DDIM with $\eta = 0$) and SDE samplers using both the simple depicted in Figure 5 (Left) and optimizing for stable diffusion in alignment with the aesthetic reward (Right), a major experiment detailed in the subsequent section 5.2.

**Understanding the Advantage of DNO with SDE-Based Sampling.** Intuitively, the better performance in optimization can be attributed to the finer-grained control over the generation process afforded by the SDE sampler compared to the ODE sampler. To formally state this intuition, we revisit the DDIM sampling algorithm in Algorithm 1. We consider the procedure of SDE-based sampling algorithm, DDIM with $\eta = 1$, as defining the noise-to-sample mapping $M_\theta(x_T, z_1, \ldots, z_T)$. An important observation is that the ODE sampling algorithm, DDIM with $\eta = 0$, can also be expressed using the same $M_\theta(x_T, z_1, \ldots, z_T)$, with the distinction that the noise $z_1, \ldots, z_T$ becomes deterministic and dependent on $x_T$, rather than sampled from a Gaussian distribution. Specifically, if we define the deterministic noise vectors as: $z_t^{\text{ODE}} \stackrel{\text{def.}}{=} \frac{\sqrt{1-\alpha_{t-1}} - \sqrt{1-\alpha_{t-1}-\sigma_t^2}}{\sigma_t} \epsilon_\theta(x_t, t)$, for $t = 1, \ldots, T$, where $\sigma_t = \sqrt{(1 - \alpha_{t-1}/(1 - \alpha_t)}\sqrt{1 - \alpha_t/\alpha_{t-1}}$, then the sampling process of DDIM with $\eta = 0$ can be expressed as $M_\theta(x_T, z_1^{\text{ODE}}, \ldots, z_T^{\text{ODE}})$. In this context, the advantage of SDE-based sampling becomes evident:

$$\underbrace{\max_{x_T, z_1, \ldots, z_T} r\left(M_\theta(x_T, z_1, \ldots, z_T)\right)}_{\text{DNO with SDE}} \geq \underbrace{\max_{x_T} r\left(M_\theta(x_T, z_1^{\text{ODE}}, \ldots, z_T^{\text{ODE}})\right)}_{\text{DNO with ODE}}, \tag{5}$$

meaning that running DNO with SDE will yield better results, or at least as good as DNO with ODE for aligning diffusion models. Based on this conclusion, our work will focus on optimizing the SDE-based sampling (DDIM with $\eta = 1$) for the remainder of the study. Additionally, we fix the number of generation steps $T$ to 50 throughout this work for simplicity.

## 3 OUT-OF-DISTRIBUTION REWARD-HACKING IN NOISE OPTIMIZATION

It has been observed that when aligning generative models (e.g., including autoregressive language models or diffusion models) with reward functions, one can experience the so-called reward-hacking, i.e., the optimized samples yield high rewards but do not possess the desirable properties (Miao et al., 2024; Chen et al., 2024a). Generally, there are two different types of reward-hacking. In the first type, the reward function admits some shortcuts, so the optimized samples score high rewards but remain barely distinguishable from the samples of the pretrained distribution. The second type is related to the generative model used – the optimized samples no longer fall within the support of the pretrained distribution after optimization. We denote this second type of reward-hacking as **Out-Of-Distribution (OOD) Reward-Hacking**. In this work, we will focus on this second type and reveal that OOD reward-hacking is a common issue in DNO. In Appendix B.2, we provide two visualized examples of the phenomenon of OOD Reward-Hacking using both a 2-dimensional diffusion model and an image diffusion model.

One of our key contributions in this work is to identify one critical cause of OOD reward-hacking in noise optimization. That is, the optimized noise vectors stray towards the low-probability regions of the high-dimensional standard Gaussian distribution; in other words, there is an extremely low chance of such noise vectors being sampled from the Gaussian distribution. As diffusion models are originally trained with Gaussian noise, when the noise vectors originate from these low-probability areas—such as vectors comprised entirely of zeros—the neural network within the diffusion models may incur significant approximation errors for these particular inputs. This error, in turn, leads to the generation of out-of-distribution samples. In the subsequent section, we introduce a novel method to measure the extent to which noise is part of the low-probability region by leveraging the classical concentration inequalities for high-dimensional Gaussian distributions.

### 3.1 QUANTIFYING LOW-PROBABILITY REGION VIA CONCENTRATION INEQUALITIES

High-dimensional Gaussian distributions possess several unique properties. For instance, it is known that the all-zero vector is the most probable in terms of the probability density function (p.d.f) of the standard Gaussian distribution. However, in practice, it is nearly impossible to obtain samples near the all-zero vector from a Gaussian distribution, as it resides within a low-probability region.

In high-dimensional statistics, concentration inequalities are usually employed to describe these distinctive properties and delineate the low-probability regions of high-dimensional distributions. In the following lemma, we present two classical inequalities for the standard Gaussian distribution.

**Lemma 1** ((Wainwright, 2019))**.** *Consider that $z_1, ..., z_m$ follow a $k$-dimensional standard Gaussian distribution. We have the following concentration inequalities for the mean and covariance:*

$$\Pr\left[\left\|\frac{1}{m}\sum_{i=1}^{m} z_i\right\| > M\right] < p_1(M) \overset{\text{def.}}{=} \max\left\{2e^{-\frac{mM^2}{2k}}, 1\right\}, \tag{6}$$

$$\Pr\left[\left\|\frac{1}{m}\sum_{i=1}^{m} z_i z_i^\top - I_k\right\| > M\right] < p_2(M) \overset{\text{def.}}{=} \max\left\{2e^{-\frac{m\left(\max\left\{\sqrt{1+M}-1-\sqrt{k/m},0\right\}\right)^2}{2}}, 1\right\}. \tag{7}$$

In practice, to determine if an $n$-dimensional vector $z$ lies within a low-probability region, we can factorize $n$ as $n = m \cdot k$, and divide $z$ into $m$ subvectors: $z = [z_1^1, ..., z_m^k]$, where $n = m \cdot k$ and $z_i = [z_i^1, ..., z_i^k] \sim \mathcal{N}(0, I_k))$. Then, we compute $M_1(z) = \left\|\frac{1}{m}\sum_{i=1}^{m} z_i\right\|$ and $M_2(z) = \left\|\frac{1}{m}\sum_{i=1}^{m} z_i z_i^\top - I_k\right\|$. Finally, we can determine that $z$ lies in a low-probability region if both $p_1(M_1(z))$ and $p_2(M_2(z))$ are low.

**Remark 2.** *According to (Wainwright, 2019), the two inequalities equation 6 and equation 7 are tight when $m/k$ is large. On the other hand, $k = 1$ is not advisable, as it examines only the mean and variance of the noise vector, but not the covariance of different subvectors. In this work, we empirically found that $k = 2$ serves as a good default choice. In Appendix F, we provide a more detailed analysis for choosing an appropriate $k$.*

An important point to note is that the standard Gaussian distribution is invariant to permutation, i.e., for any permutation matrix $\Pi$, if $z$ follows a standard Gaussian distribution, the permuted vector $\Pi z$ will have the same probability behavior. With this insight, to increase the robustness of the probability measure $p_1$ and $p_2$, a natural idea is to examine the probability of many permuted vectors. Specifically, given $q$ permutation matrices $\Pi_1, ..., \Pi_q$, we define the following indicator metric,

$$P(z) \overset{\text{def.}}{=} \min\left\{\min_{i\in\{1,...,q\}} p_1\left(M_1(\Pi_i z)\right), \min_{i\in\{1,...,q\}} p_2\left(M_2(\Pi_i z)\right)\right\}. \tag{8}$$

**Interpretation of $P(z)$.** If the probability $P(z)$ is low, it implies that there exists a permutation matrix $\Pi_i$ such that the noise vector $\Pi_i z$ is in the low-probability region of the standard Gaussian distribution. Therefore, due to the permutation-invariant property, the noise vector $z$ is also less likely to be sampled from the standard Gaussian distribution. In practice, we utilize randomly generated permutation matrices and have found that setting $q = 100$ results in empirically good performance. In Appendix D, we provide some visualized empirical evidence to show that $P(z)$ serves as a good indicator for determining if the generated samples are OOD.

### 3.2 Probability-Regularized Noise Optimization

With the insights discussed above, a natural idea for preventing OOD reward hacking is to regularize noise vectors to remain within the high-probability region of the Gaussian distribution. To achieve this, we propose the following **Probability-Regularized Noise Optimization** problem:

$$\max_z r(M_\theta(z)) + \gamma \mathbb{E}_\Pi\left[\log p_1(M_1(\Pi z)) + \log p_2(M_2(\Pi z))\right], \tag{9}$$

where $\gamma$ is the coefficient used to control the regularization effect. In particular, for the regularization term, we use the expectation of the log probabilities over the permutation matrices, rather than the minimum probability $P(z)$. This is because the expectation is smoother for optimization purposes.

## 4 Tackling Non-Differentiable Reward Functions

In the previous section, DNO method has been applied to optimize *differentiable* reward functions. However, in many applications the ground-truth gradient of the reward function is unavailable. Such non-differentiable properties can arise from various scenarios; here we present two representative cases. Firstly, the reward may be computed through simulation-based procedures, such as the JPEG-compressibility employed in DDPO (Black et al., 2023), which calculates the size of an image in bits after running a JPEG compression algorithm. Additionally, the reward function itself may be a black box provided through online API providers, as in the setting considered in (Sun et al., 2022). This scenario is common when the reward function is a large neural network model, like those in (Wang et al., 2024a; Lin et al., 2024), making it impossible to directly obtain the gradient

for optimization. To address these scenarios, we explore adapting the noise optimization approach to handle the optimization of non-differentiable reward functions by estimating the gradient with function values. Specifically, we explore three methods under this setting.

**Method 1.** Concerning optimization with only function value, a major family of optimization approaches is zeroth-order optimization algorithms, including ZO-SGD (Nesterov & Spokoiny, 2017). This method treats the entire mapping $r \circ M_\theta(\cdot)$ as a black-box function and seeks to estimate the gradient of $r \circ M_\theta(\cdot)$ via function value queries.

**Method 2.** It is worth noting that for the mapping $r \circ M_\theta(\cdot)$, only the gradient of the reward function $r(x)$ is not available, and we are still able to compute the gradient of $M_\theta(z)$. Therefore, a straightforward idea is to adopt a hybrid gradient approach—only to estimate the gradient of $r$, while using the ground truth gradient for $M_\theta(z)$. Specifically, we denote that the initial noise is $z$, and the generated sample is $x = M_\theta(z)$. Firstly, we can estimate the gradient of $\nabla r(x)$ in a similar fashion with the ZO-SGD (Nesterov & Spokoiny, 2017):

$$H_1(x) = \mathbb{E}_{\xi \sim \mathcal{N}(0,I)} \left[ (r(x + \mu\xi) - r(x))\, \xi \right] \approx C_1 \nabla r(x), \tag{10}$$

where $\mu$ is the coefficient for perturbation, and $C_1$ is some constant. With the estimated gradient $H_1(x)$ for the reward function $r(x)$, we can use the following estimated gradient $G_1(z)$ for optimization:

$$G_1(z) \stackrel{\text{def.}}{=} H_1(x) \cdot \nabla_z M_\theta(z) \approx C_1 \nabla_z r \circ M_\theta(z), \tag{11}$$

where the main idea is to replace the ground truth $\nabla r(x)$ in the chain-rule of differentiating $r(M_\theta(z))$. We refer this method as **Hybrid-1** in the following sections.

**Remark 3.** *As one can observe, the computation of equation 11 involves the Jacobian $\nabla_z M_\theta(z)$. However, it is important to note that when we only require the vector-Jacobian product $H_1(x) \cdot \nabla_z M_\theta(z)$, it is unnecessary to compute the full Jacobian $\nabla_z M_\theta(z)$. In Appendix E, we describe an elegant and efficient way to implement equation 11 using an auto-differentiation technique.*

**Method 3.** There is a crucial drawback in the gradient estimator equation 10, that one needs to query the reward function $r(\cdot)$ with noisy input $x + \mu\xi$. When the reward function is only defined on some manifold $\mathcal{M}$, e.g., defined on the image manifold, rather than the whole space $\mathbb{R}^n$, this can lead to severe problems, because, for some $x \in \mathcal{M}$, the noisy sample $x + \mu\xi$ may no longer stay within $\mathcal{M}$. To remedy this issue, we propose to perturb the sample through the latent noise, rather than directly in the sample space. Specifically, our proposed new gradient estimator for $\nabla r(x)$ is

$$H_2(x) = \mathbb{E}_{\xi \sim \mathcal{N}(0,I)} \left[ (r(M_\theta(z + \mu\xi)) - r(x)) (M_\theta(z + \mu\xi) - x) \right]. \tag{12}$$

Following a similar proof in (Nesterov & Spokoiny, 2017), we can also show that $H_2(x) \approx C_2 \nabla r(x)$ for some constants $C_2$. As we can see, when computing the gradient equation 12, we ensure that we query the reward function $r(\cdot)$ with only samples that are within the manifold of the pretrained distribution. Similar to Hybrid-1, we can adopt a gradient estimator $G_2(z)$ with $H_2(x)$:

$$G_2(z) \stackrel{\text{def.}}{=} H_2(x) \cdot \nabla_z M_\theta(z). \tag{13}$$

We refer this method equation 13 as **Hybrid-2**. As we will see in Section 5.3, this Hybrid-2 method is significantly faster than the other two in terms of optimization speed.

## 5 EXPERIMENTS

In this section, we aim to demonstrate the effectiveness of the method proposed above. For all subsequent experiments, we utilize Stable Diffusion v1.5 (Rombach et al., 2022) as the base model for noise optimization. For each figure, we perform the optimization using 1,000 different random seeds and report the average value along with the standard deviation (std) of the results. For comprehensive details regarding the implementation of our proposed methods, as well as information on hyperparameters, we refer readers to Appendices E and F. Additionally, we provide examples to visualize the optimization process in Appendix B. For all the following experiments, unless explicitly stated otherwise, a single run of DNO is performed on a single A800 GPU.

### 5.1 EXPERIMENTS ON IMAGE BRIGHTNESS AND DARKNESS REWARD FUNCTIONS

**Experiment Design.** In this section, we design an experiment to demonstrate the effectiveness of DNO as described in Section 2, and the probability regularization proposed in Section 3.2. We consider two settings: The first involves optimizing the brightness reward, which is the average value of all pixels—the higher this value, the brighter the image becomes—with the prompt "black <animal>", where the token <animal> is randomly selected from a list of animals. The second

setting involves optimizing the darkness reward, defined as the negative of the brightness reward, with the prompt "white <animal>". The primary rationale behind designing such experiments is the inherent contradiction between the prompt and the reward, which makes it easier to trigger the OOD reward-hacking phenomenon. Moreover, it is straightforward to verify whether the generated samples are out-of-distribution by simply examining the color of the generated animals. In these experiments, we compare the noise optimization process with and without probability regularization to assess the capability of probability regularization in preventing the OOD reward-hacking phenomenon.

**Importance of the Brightness and Darkness Reward Functions.** While the primary purpose of using these two reward functions is to better examine the effectiveness of probability regularization, it is also important to highlight their practical utility. There is often a genuine need to generate images with extremely dark or bright backgrounds, which cannot be achieved by the base models through prompting alone, as reported in the notable research by (CrossLabs, 2023).

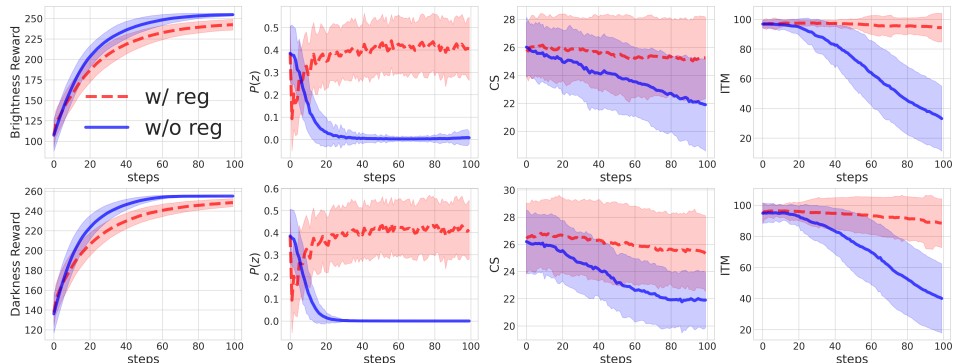

Figure 2: Comparison of DNO with and without probability regularization. Upper row: Optimizing for the brightness reward, i.e., the average value of all pixels in the images. Lower row: Optimizing for the darkness reward, i.e., the negative of the brightness reward. The x-axis refers to the number of gradient ascent steps during optimization.

**Measuring the Degree of OOD.** As observed in this experiment, the reward function contradicts the input prompt, leading to an inconsistency between the generated samples and the prompt. Therefore, we utilize the CLIP Score (CS) (Radford et al., 2021), a commonly used metric for measuring the semantic similarity between images and text descriptions, to gauge the degree of OOD for the generated samples. A higher CS indicates that the sample is less likely to be out-of-distribution (OOD). In addition to the CS, we also use an MLLM-based score to measure the degree of OOD, specifically employing the Image-Text Matching (ITM) score from (Wang et al., 2024a) as the metric.

**Results.** In Figure 2, we first observe that adding the regularization term leads to a mildly slower optimization process. However, the generated samples are much more consistent with the prompt throughout the entire optimization process, as reflected by the CS and ITM curves. The trajectory of $P(z)$ further corroborates that it is a good indicator of the OOD phenomenon, as it is positively associated with CS and ITM. We also provide visualized examples in Appendix B.3, which also confirm that the probability regularization proposed in Section 3.2 can effectively prevent the generated samples from becoming OOD. We also examined the performance of other inference-time methods, LGD and Best-of-N (BoN) under this setting, see Appendix E.2.

**Generating Images with Purely Bright/Dark Backgrounds.** From the images in Figure 7 of Appendix B.3, we observe that DNO with regularization can lead to images with purely dark or bright backgrounds. It is worth noting that this is a remarkable result, as DNO requires no fine-tuning. Notably, as discussed in (CrossLabs, 2023), such an effect can only be achieved by fine-tuning the diffusion models using a technique called Offset-Noise.

## 5.2 BENCHMARKING ON THREE HUMAN-ALIGNED REWARD FUNCTIONS

**Setting.** In this section, we investigate the performance of the proposed method using three common reward functions trained from human feedback data, specifically Aesthetic Score (Schuhmann et al., 2022b), HPS-v2 score (Wu et al., 2023), and PickScore (Kirstain et al., 2023), respectively. In this experiment, we also compare noise optimization with and without probability regularization. However, compared to the reward function used in the previous section, using these reward functions presents a lesser chance for the optimized image to be OOD. In this case, to measure the benefit of

| Method | SD v1.5 2s | LGD n=10 | LGD n=100 | SPIN ∼ 20h | DDPO ∼ 56h | AlignProp ∼ 12h | DNO (This work) 1 min | DNO (This work) 3 min | DNO (This work) 5 min | BoN 10 min |
|--------|-----------|----------|-----------|------------|------------|-----------------|------|------|------|------|
| Aesthetic ↑ | 5.367 | 5.340 | 5.224 | 6.248 | 7.180 | 8.940 | 5.754 | 7.202 | 8.587 | 6.531 |
| HPS ↑ | 0.278 | 0.276 | 0.271 | 0.276 | 0.287 | 0.330 | 0.285 | 0.303 | 0.324 | 0.298 |
| PickScore ↑ | 21.11 | 21.01 | 21.09 | 22.00 | / | / | 21.25 | 23.17 | 25.13 | 22.09 |

Table 1: Performance comparison. For SD v1.5 and DNO, we annotate the generation time below the name. For LGD, we annotate the number of samples used for Monte Carlo approximation. For SPIN, DDPO, and AlignProp, we annotate the estimated time for fine-tuning. All time costs in the table are measured with respect to the GPU time on a single A800 GPU. Baselines: LGD (Song et al., 2023a), SPIN (Yuan et al., 2024), DDPO (Black et al., 2023), AlignProp (Prabhudesai et al., 2023), Best-of-N (BoN).

probability regularization in maintaining the quality of the generated sample, we consider using the other two reward functions as test metrics when optimizing one of them.

**Results.** Firstly, we observe that the effect of probability regularization is less pronounced than that in Section 5.1. This observation is also reflected by our proposed indicator $P(z)$; if no regularization is applied, the value of $P(z)$ in Figure 3 decreases much slower than that in Figure 2. Nonetheless, by adding the regularization term to noise optimization, we can stabilize the value of $P(z)$ throughout the optimization process and also improve the test metrics. For instance, when optimizing the aesthetic reward, the regularization has no significant effect on the optimization speed, while it prevents the test metrics, i.e., the HPS score and Pick Score, from decreasing throughout the process.

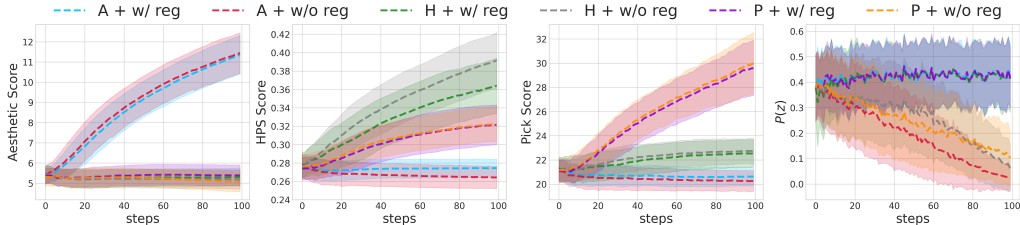

Figure 3: Comparison of running DNO with three human-like reward functions, with and without regularization. When optimizing one reward function, the other two are used as test metrics. A, H, P are short for Aesthetic Score, HPS Score, and Pick Score, respectively. The name for each line comprises the used reward function and whether the regularization is used. For example, A + w/ reg means optimizing aesthetic score with regularization.

**Comparison to Existing Alignment Methods.** We summarize the performance of DNO with probability regularization from Figure 3 into Table 1 and compare it to the major existing alignment methods discussed in the introduction. As shown, the performance of DNO matches that of state-of-the-art tuning-based alignment methods without any fine-tuning on the network models, all within a reasonable time budget for generation. More importantly, we demonstrate that DNO provides an worthwhile trade-off between inference time and the reward of the generated samples. On one hand, another inference-time method, LGD (Song et al., 2023a), performs poorly with these complex reward functions, as it is impossible to estimate the gradient of the reward functions without a complete generation process. On the other hand, we also examined the most fundamental inference-time alignment algorithm, Best-of-N (BoN) Sampling, which generates N samples and selects the one with the highest reward. In this experiment, we fix the time budget for BoN to 10 minutes, and we observe that DNO outperforms it by a large margin, demonstrating that DNO presents a highly advantageous trade-off between inference time and reward. As a complement to Table 1, we also provide the numerical results for the test metrics of our DNO and the tuning-based method AlignProp in Appendix E.2, analyzing their performance on the OOD reward-hacking issue.

**Using Fewer Steps to Reduce Inference Time.** In Figure 3, we show that our proposed DNO can achieve high reward values within approximately 3–5 minutes. While this already represents a highly advantageous performance, 3–5 minutes of optimization may still be prohibitively long in prac-

| $T$ | 10 | 15 | 20 | 25 | 50 |
|-----|-----|-----|-----|-----|-----|
| Aesthetic ↑ | 6.992 | 7.496 | 6.773 | 6.381 | 5.754 |
| HPS ↑ | 0.342 | 0.341 | 0.306 | 0.293 | 0.285 |
| PickScore ↑ | 23.98 | 24.82 | 23.69 | 23.02 | 21.25 |

Table 2: Running DNO with different numbers of diffusion steps $T$, while fixing the time budget to 1 minute.

tice. However, we note that reducing the number of diffusion steps can significantly decrease

optimization time. In Table 1, we fix the diffusion steps to 50, as this is the setting used by all other algorithms to ensure a fair comparison. In practice, 50 diffusion steps is a fairly long choice, and using 15–25 steps can still result in sufficiently good samples. Therefore, in Table 2, we present the performance of DNO with different numbers of denoising steps, fixing the time budget to 1 minute, as well as other hyperparameters like learning rate and regularization coefficient. As Table 2 shows, with 15 diffusion steps, DNO can achieve high reward values within 1 minute.

Finally, we would like to highlight that this alignment process can run with memory usage of less than **15GB**, and thus can easily fit into just **one consumer-level GPU**. In contrast, current tuning-based methods require significantly more computing resources, typically 4-8 advanced GPUs like A100 (Black et al., 2023; Prabhudesai et al., 2023).

## 5.3 Optimization for Non-Differentiable Reward Functions

**Setting.** In this section, we aim to explore the three extensions proposed in Section 4 for handling non-differentiable reward functions. For Method 1, we use ZO-SGD (Nesterov & Spokoiny, 2017), and compare it with our proposed Hybrid-1 and Hybrid-2. We consider using two reward functions: The Jpeg Compressibility score, which is the file size of the image after compressing it using the JPEG algorithm. This reward function was also used in (Black et al., 2023) and is intrinsically non-differentiable. The second reward function is the Aesthetic Score used in the previous section. Unlike previous experiments, in this section we treat the Aesthetic Score as a non-differentiable reward function for optimization. The goal is to simulate a scenario where the neural network model of the Aesthetic Score can only be queried via an API provider that returns the score rather than its gradient. We compare the three methods in terms of optimization steps and the final score of the reward function. For ease of demonstration, we do not add the probability regularization term to the optimization process. Moreover, it is important to note that in these three methods, the major time expenditure comes from estimating the gradient with finite samples. For a fair comparison among these methods, we set the number of samples for estimating the gradient separately so that the total time spent estimating the gradient is the same for all three methods.

**Results.** The main results are visualized in Figure 4. Initially, we observe that the Hybrid-2 method is significantly faster than the other two methods, and the final score is also higher for both reward functions used in this experiment. Furthermore, in both scenarios, the ZO-SGD method exhibits the slowest optimization speed. Another interesting finding is the poor performance of Hybrid-1 in optimizing the Aesthetic Reward. This mainly occurs because the aesthetic reward function can only work with image inputs, which validates our initiative to propose the Hybrid-2 method in Section 4. Finally, when comparing the optimization of the aesthetic score in Figures 3 and 4, we can also note

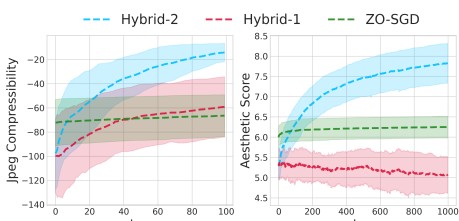

Figure 4: Comparing three methods on two reward functions. For better visualization purposes, when plotting the line for ZO-SGD, we compute and plot the current best reward instead of the current reward due to the extremely high variance.

that using the true gradient results in substantially fewer steps than using the estimated gradient.

## 6 Conclusions

In this work, we present a comprehensive study on Direct Noise Optimization (DNO) for aligning diffusion generative models at inference-time. We introduce variants of DNO designed to efficiently address challenges such as out-of-distribution reward-hacking and the optimization of non-differentiable reward functions. More significantly, we demonstrate the exceptional efficacy of DNO, underscoring its capacity to rival tuning-based methods. The primary limitation of DNO lies in its integration with the sampling process of diffusion models, leading to a substantial increase in processing time compared to direct sampling. Nonetheless, we argue that the additional time cost, being within a reasonable and acceptable range, is a worthwhile trade-off for attaining high-reward generated samples in a wide range of real-world applications. We anticipate DNO gaining greater attention in future research and applications due to its flexibility to accommodate any reward function—or even a combination of different reward functions—while demanding only modest computing resources, which positions it as an accessible tool for many practitioners.

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

# APPENDIX

## A    DDIM SAMPLING ALGORITHM

In Algorithm 1 described below, we summarize the sampling algorithm for diffusion models, DDIM (Song et al., 2020a), which is essentially the Euler method for solving ODEs/SDEs. The diffusion coefficients $\alpha_1, \ldots, \alpha_T$ in Algorithm 1 are computed using the coefficient functions $f(t)$ and $g(t)$, with the detailed computation found in (Song et al., 2020b; Karras et al., 2022). The coefficient $\eta$ in DDIM determines whether we are solving the ODE or the SDE, with $\eta = 0$ corresponding to ODE and $\eta > 0$ corresponding to SDE.

---

**Algorithm 1** DDIM Sampling Algorithm

---

**Require:** Discretization timesteps $T$, diffusion coefficient $\alpha_1, ..., \alpha_t$, initial noise $x_T \sim \mathcal{N}(0, I)$, noise vectors $z_1, ..., z_T \sim \mathcal{N}(0, I)$, coefficient $\eta \in [0, 1]$ for balancing ODE and SDE, learned score network $\epsilon_\theta(\cdot, \cdot)$.
1: **for** $t = T$ to 1 **do**
2:      Compute $\sigma_t = \eta \sqrt{(1 - \alpha_{t-1}/(1 - \alpha_t)}\sqrt{1 - \alpha_t/\alpha_{t-1}}$.
3:      $x_{t-1} = \sqrt{\alpha_{t-1}/\alpha_t} \cdot x_t - \left( \sqrt{\alpha_{t-1}(1 - \alpha_t)/\alpha_t} - \sqrt{1 - \alpha_{t-1} - \sigma_t^2} \right) \epsilon_\theta(x_t, t) + \sigma_t z_t$
4: **end for**
5: **return** $x_0$

---

## B    VISUALIZATION

To assist the reader in understanding the optimization process of DNO and also to provide a qualitative evaluation, we offer several visualizations in this section.

### B.1    A SIMPLE EXAMPLE

In this section, we present a simple example to visualize the process of noise optimization. Specifically, we trained a toy diffusion model for generating uniform distribution on a ring with a radius between 0.8 and 1.2, and the initial distribution is visualized in Figure 5a, where each red point denotes a single sample drawn from the trained diffusion model. We then solve the noise optimization problem equation 3 with the reward function $r(x) = \sin(4\pi x[1]) + \sin(4\pi x[2]) - \left( (x[1] - 1)^2 - x[2]^2 \right) / 5$, a highly nonconvex function with many local maxima. To perform this optimization, we solve the DNO problem equation 3 using gradient ascent with the learning rate set to 0.01. In Figures 5b, 5c, and 5d, we visualize the optimized samples after 10, 50, and 100 gradient steps, respectively. We can observe that the distribution of the samples shifts toward a distribution on the local maxima of the reward function.

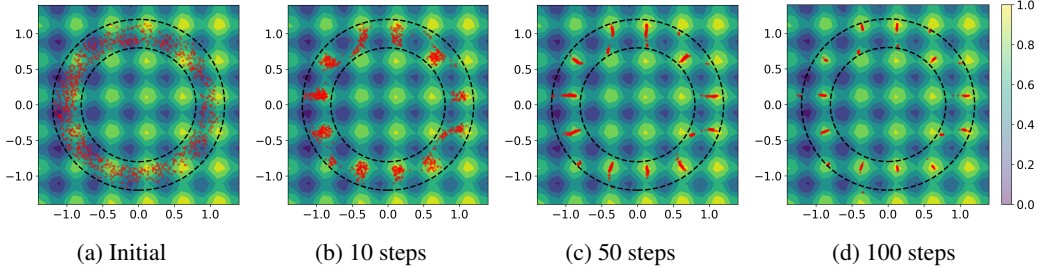

     (a) Initial          (b) 10 steps          (c) 50 steps          (d) 100 steps

Figure 5: Example 1: Evolution of the sample distribution of a toy diffusion model while running DNO to maximize a non-convex reward function.

### B.2    SIMPLE EXAMPLES FOR OOD REWARD-HACKING

In this section, we present two examples using both a simple diffusion model from the Example 1 described above and the open-source image diffusion model SD v1.5 (Rombach et al., 2022). For the first example, we revisit the pretrained distribution displayed in Figure 5, modifying the reward function to $r(x) = -(x[1] - 1.4)^2 - (x[2] - 1.4)^2$. Figure 6a exhibits the pretrained distribution of the diffusion models, where we note that every sample stays within the support. However, as illustrated in Figure 6b, after 1000 gradient steps of optimization for the reward function, all the generated samples become out-of-distribution. In our second example, we examine optimization for the "brightness" reward—specifically, the average value of all pixels in an image—using SD v1.5 as the image diffusion model. We start with the prompt "black duck", with the initial image depicted in Figure 6c. After 50 gradient steps of optimization for the brightness reward, it becomes apparent that the generated samples diverge from the original prompt "black duck" and transform into a "white duck", as evidenced in Figure 6d. Ideally, in the absence of reward-hacking, the generated samples should always adhere to the "black duck" prompt while incorporating the overall brightness in the images.

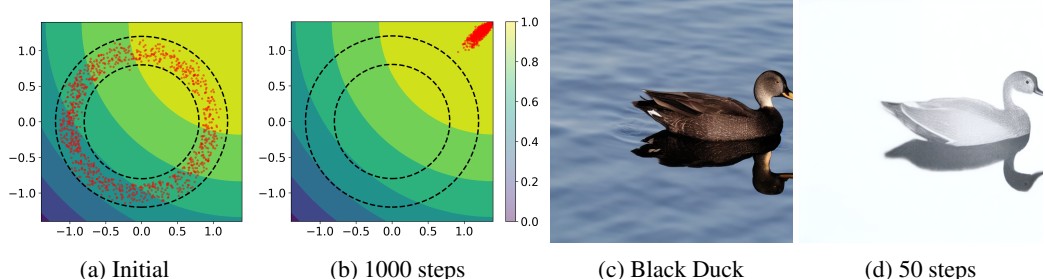

(a) Initial      (b) 1000 steps      (c) Black Duck      (d) 50 steps

Figure 6: Examples of OOD Reward-Hacking

### B.3 COMPARING WITH AND WITHOUT REGULARIZATION

In this section, our objective is to demonstrate the impact of the probability regularization term on the optimization process through visual examples. Specifically, we present examples from three settings. The first two examples are derived from the experiments in Section 5.1, while the last example is from the experiment on optimizing the aesthetic score in Section 5.2.

The examples can be seen in Figure 7. As observed across all examples, the optimization process that integrates the regularization term consistently prevents the generated samples from falling into the category of being Out-Of-Distribution (OOD).

### B.4 QUALITATIVE EXAMPLES

In this section, we aim to provide additional visualized examples for our proposed method.

Firstly, in Figure 8, we present examples from the optimization of all three popular human-level reward functions discussed in Section 5.2. As can be observed, the optimization process indeed results in an increase in human preference throughout.

Furthermore, we also include examples from the experiments in Section 5.3, that is, optimizing JPEG Compressibility Score and Aesthetic Score using the Hybrid-2 method for non-differentiable optimization. These examples in Figure 9 effectively showcase the efficiency of Hybrid-2 in both estimating the gradient and optimizing.

We also present some non-cherry-picked examples of aligning Stable Diffusion XL (Podell et al., 2024) with DNO across four reward functions and four popular prompts from Reddit, see Figure 10. Note that this effect is achieved **without fine-tuning the diffusion models.** The experiment was conducted on a single A800 GPU. , and also the setting for Figure 10. In these examples, we only use the base model of SDXL (Podell et al., 2024) as the image diffusion model. We adopt the DDIM sampler with 50 steps and $\eta = 1$ for generation, and optimize all the injected noise in the generation process, the same as most experiments in this work. The classifier-free guidance is set to 5.0. For each reward function, we adopt the same hyperparameters for the optimizer and regularization terms as the experiments in Section 5.1 and 5.2. From top to bottom in Figure 10, the used prompts are listed as follows:

1. *dark alley, night, moon Cinematic light, intricate detail, high detail, sharp focus, smooth, aesthetic, extremely detailed*

2. *1970s baseball player, hyperdetailed, soft light, sharp, best quality, masterpiece, realistic, Canon EOS R3, 20 megapixels.*

3. *a rabbit, wildlife photography, photograph, high quality, wildlife, f 1.8, soft focus, 8k, national geographic, award - winning photograph by nick nichols.*

4. *A beef steak, depth of field, bokeh, soft light, by Yasmin Albatoul, Harry Fayt, centered, extremely detailed, Nikon D850, award winning photography*

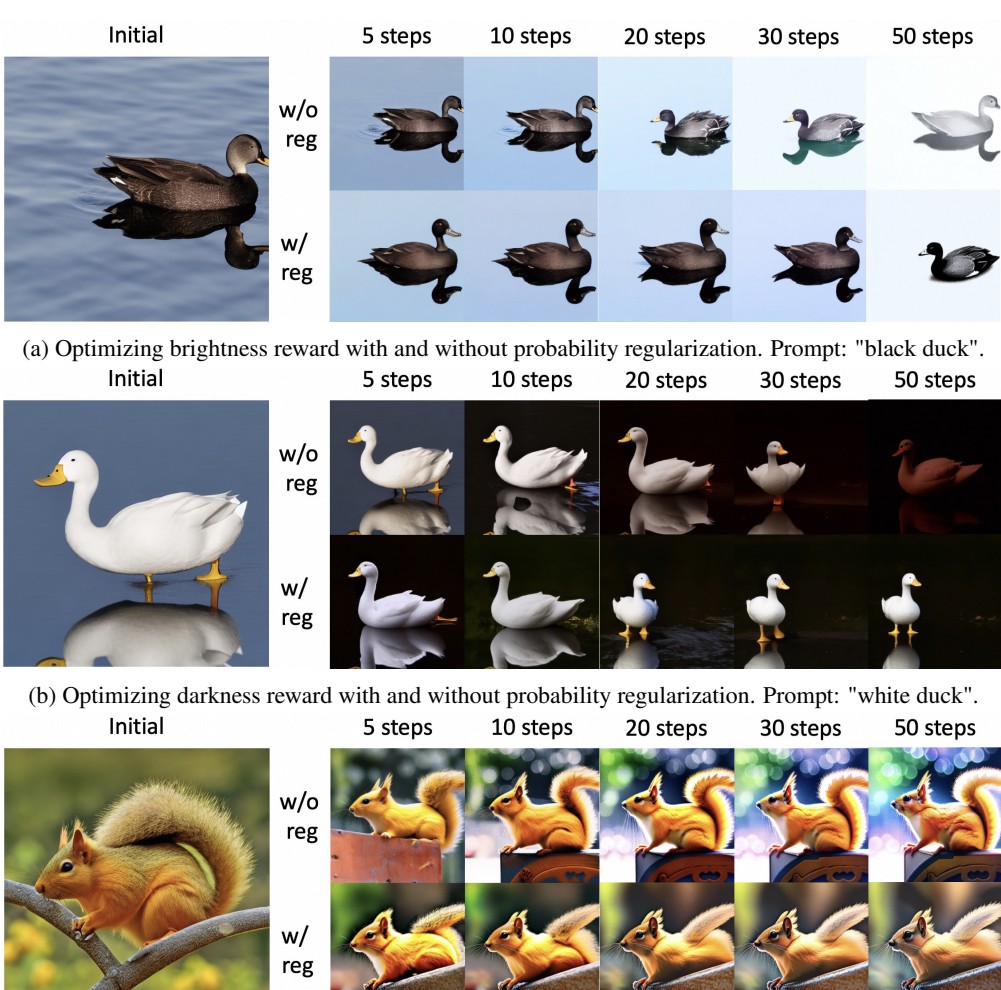

(a) Optimizing brightness reward with and without probability regularization. Prompt: "black duck".

(b) Optimizing darkness reward with and without probability regularization. Prompt: "white duck".

(c) Optimizing Aesthetic Score with and without probability regularization. Prompt: "yellow squirrel"

Figure 7: Examples of optimized samples with and without regularization

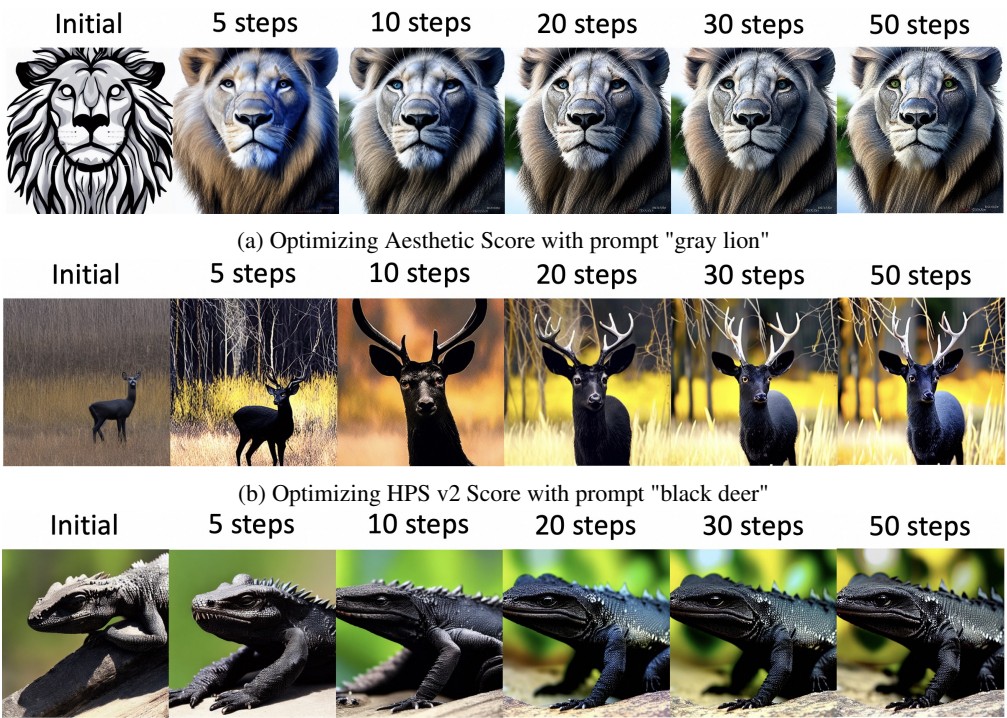

(a) Optimizing Aesthetic Score with prompt "gray lion"

(b) Optimizing HPS v2 Score with prompt "black deer"

(c) Optimizing PickScore with prompt "black lizard"

Figure 8: Representative examples of optimizing reward functions trained on human feedback data.

### B.5 VISUALIZATION OF DNO

We provide a visual illustration in Figure 11 to describe the main procedure for DNO using the DDIM sampling procedure detailed in Algorithm 1. As shown, DNO, similar to LGD (Song et al., 2023a), operates at inference-time and does not require tuning the network parameter $\theta$. However, it requires more time for generation compared to direct sampling, as the optimization is integrated with the sampling process, meaning the optimization is performed for each new sample generated. Despite this, as we will demonstrate in Section 5, the extra time needed for the DNO approach is a worthwhile trade-off for obtaining high-reward samples.

## C THEORETICAL RESULTS

*Proof for Theorem 1.* To leverage the $L$-smoothness assumption, we need to state a classical lemma for smooth optimization.

**Lemma 2** (Descent Lemma (Bertsekas, 1997))**.** *For any $z_1$ and $z_2$, we have*

$$r \circ M_\theta(z_2) \geq r \circ M_\theta(z_1) + \nabla r \circ M_\theta(z_1) \cdot (z_2 - z_1) - \frac{L}{2} \|z_2 - z_1\|_2^2. \qquad (14)$$

Now for any $z$ and steps $t \geq 1$, with the descent lemma, we have

$$r \circ M_\theta(g_{t+1}(z)) \geq r \circ M_\theta(g_t(z)) + \nabla r \circ M_\theta(g_t(z)) \cdot (g_{t+1}(z) - g_t(z)) - \frac{L}{2} \|g_{t+1}(z) - g_t(z)\|_2^2.$$

Notice that by the definition of $g_t \cdot$ we have

$$g_{t+1}(z) - g_t(z) = g_t(z) + \ell \nabla r \circ M_\theta(g_t(z)) - g_t(z)$$
$$= \ell \nabla r \circ M_\theta(g_t(z)).$$

Therefore, we have

$$r \circ M_\theta(g_{t+1}(z)) \geq r \circ M_\theta(g_t(z)) + \left( \ell - \frac{\ell^2 L}{2} \right) \|\nabla r \circ M_\theta(g_t(z))\|_2^2.$$

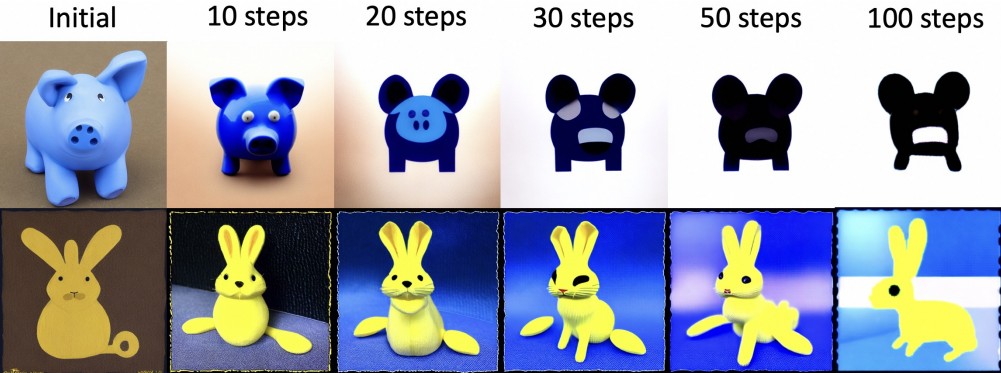

(a) Optimizing Jpeg Compressiblity with Hybrid-2 gradient approximation. Upper: prompt "blue pig". Lower: prompt "yellow rabbit".

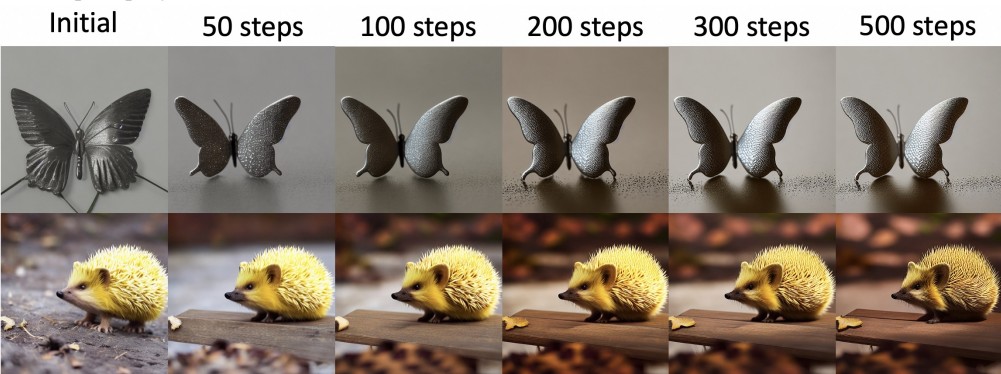

(b) Optimizing Aesthetic Score with Hybrid-2 gradient approximation. Upper: prompt "silver butterfly". Lower: prompt "yellow hedgehog".

Figure 9: Representative examples for non-differentiable optimization

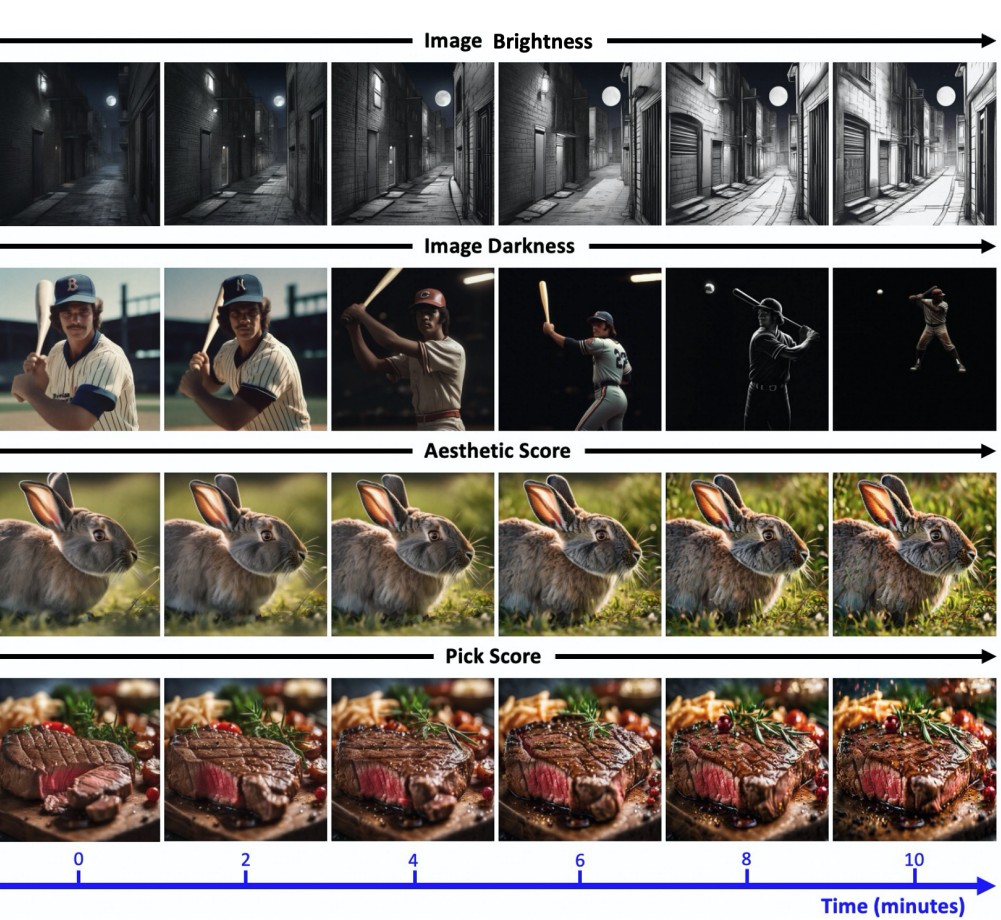

Figure 10: Running DNO with SDXL

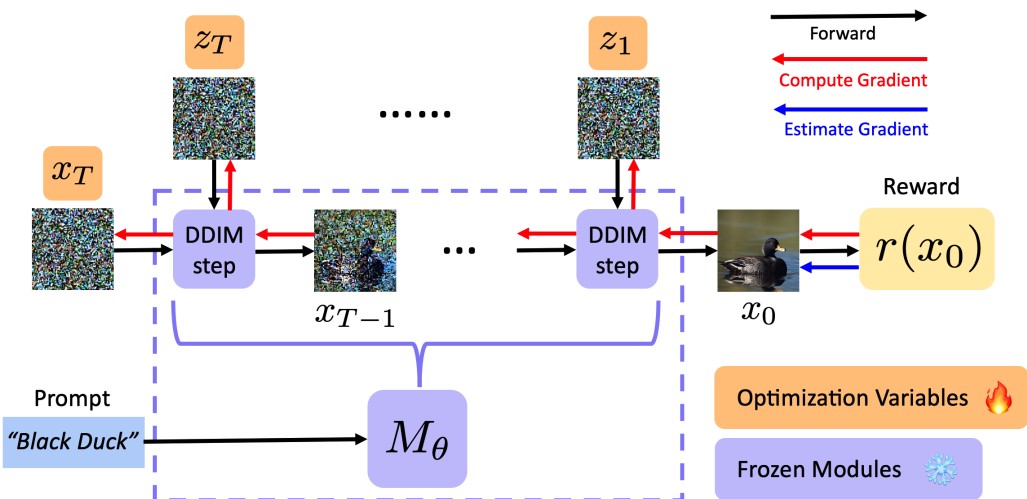

Figure 11: Overview of the DNO procedure with the DDIM sampling algorithm: DNO seeks to optimize only those Gaussian noise vectors $\{x_T, z_1, z_2..., z_T\}$ to maximize the reward value of a single generated sample $x_0$. To facilitate the gradient backpropagation from $x_0$ to $\{x_T, z_1, z_2..., z_T\}$, we leverage the technique of *gradient checkpointing*. It is worth noting that when using $\eta = 0$ for DDIM sampling, there is no need to compute the gradient for $z_1, ..., z_T$, as the generated sample $x_0$ depends exclusively on $x_T$. When computing the gradient from $r(x_0)$ to $x_0$, we can use either ground-truth gradient $\nabla r$ or an estimated gradient $\widehat{\nabla r}$, depending on whether the reward function $r(\cdot)$ is differentiable.

Taking the expectation over $z \sim \mathcal{N}(0, I)$ we have

$$\mathbb{E}_{z \sim \mathcal{N}(0,I)} \left[ r \circ M_\theta(g_{t+1}(z)) \right] \geq \mathbb{E}_{z \sim \mathcal{N}(0,I)} \left[ r \circ M_\theta(g_t(z)) \right] +$$
$$\left( \ell - \frac{\ell^2 L}{2} \right) \mathbb{E}_{z \sim \mathcal{N}(0,I)} \left[ \| \nabla r \circ M_\theta(g_t(z)) \|_2^2 \right].$$

By using the change of variable formula for distribution, we can easily see that

$$\mathbb{E}_{x \sim p_{t+1}(x)} r(x) = \mathbb{E}_{z \sim \mathcal{N}(0,I)} \left[ r \circ M_\theta(g_{t+1}(z)) \right],$$

and

$$\mathbb{E}_{x \sim p_t(x)} r(x) = \mathbb{E}_{z \sim \mathcal{N}(0,I)} \left[ r \circ M_\theta(g_t(z)) \right],$$

Therefore, we conclude with

$$\mathbb{E}_{x \sim p_{t+1}(x)} r(x) \geq \mathbb{E}_{x \sim p_t(x)} r(x) + \left( \ell - \frac{\ell^2 L}{2} \right) \mathbb{E}_{z_0 \sim N(0,I)} \left\| \nabla_z r \circ M_\theta(z)_{|z=g_t(z_0)} \right\|_2^2 \quad (15)$$
$$\geq \mathbb{E}_{x \sim p_t(x)} r(x).$$

$\square$

**When Does the Distribution Stop Improving?** As observed, the distribution ceases to improve when the second term in Equation equation 15 becomes zero. Initially, we note that the optimized distribution stops improving when $\mathbb{E}_{z_0 \sim N(0,I)} \left\| \nabla_z r \circ M_\theta(z)_{|z=g_t(z_0)} \right\|_2^2 = 0$. In statistical terms, this implies that $\nabla_z r \circ M_\theta(z)_{|z=g_t(z_0)}$ is a zero vector with probability one.

To discern the circumstances under which this zero vector occurs, let us assume that $z_0$ is some fixed noise vector and consider the scenario where

$$\nabla_x r(x)_{|x=M_\theta(g_t(z_0))} \cdot \nabla_z M_\theta(z)_{|z=g_t(z_0)} = \vec{0}, \quad (16)$$

Here, we denote $G_1(z_0) = \nabla_x r(x)_{|x=M_\theta(g_t(z_0))}$ as the gradient of the reward functions on the generated sample, and $G_2(z_0) = \nabla_z M_\theta(z)_{|z=g_t(z_0)}$ representing the Jacobian matrix of the noise-to-sample mapping. We categorize the situation in Equation equation 16 into three cases:

**Type-I:** $\|G_1(z_0)\| = 0$ and $\|G_2(z_0)\| > 0$. Here, the gradient of the reward function on the generated sample is zero, indicating that the generated sample has reached a stationary point (or local solution) of the reward function.

**Type-II:** $\|G_2(z_0)\| = 0$ and $\|G_1(z_0)\| > 0$. This indicates that the Jacobian matrix of the noise-to-sample mapping is zero, which often suggests that the generated sample is at the boundary of the support of the distribution, as a zero Jacobian means that changes in the noise will not affect the generated sample.

**Type-III:** $\|G_1(z_0)\| > 0$ and $\|G_2(z_0)\| > 0$, but $\|G_1(z_0) \cdot G_2(z_0)\| = 0$. In this scenario, the gradient of the reward function on the generated sample is orthogonal to the Jacobian matrix of the noise-to-sample mapping.

In summary, the distribution will halt its improvement after the $t$-th step if it almost surely holds that $z_0$ corresponds to a Type-I, Type-II, or Type-III noise vector.

We provide examples for the three scenarios, respectively, in the following figures. First, in Figure 12a, we display examples of Type-I and Type-II by reutilizing the experiment from Figure 5. To determine the type of the noise vector, we empirically compute $\|G_1(z_0)\|$, $\|G_2(z_0)\|$, and $\|G_1(z_0) \cdot G_2(z_0)\|$ for each noise vector.

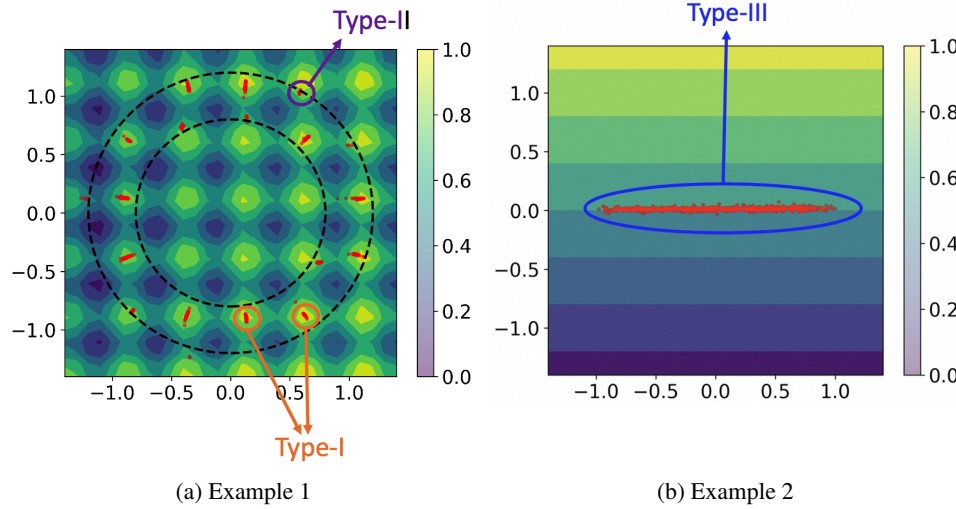

(a) Example 1          (b) Example 2

Figure 12: Examples of generated samples with Type-I, Type-II and Type-III noise vectors in the toy examples.

To showcase an example of Type-III noise vectors, we introduce a new toy example illustrated in Figure 12b. Specifically, the ground-truth distribution learned by diffusion models is uniform across a horizontal line spanning from $(-1, 0)$ to $(1, 0)$. The reward function is defined as $r(x, y) = y$. It can be readily confirmed that, for every point on this line, the gradient of the reward function is orthogonal to the Jacobian matrix of the noise-to-sample mapping. Consequently, all points along the line segment $[(-1, 0), (1, 0)]$ qualify as Type-III noise vectors.

## D    EMPIRICAL INVESTIGATION OF $P(z)$

In this section, we provide several empirical evidence to demonstrate that $P(z)$ acts as an effective indicator for the out-of-distribution phenomenon.

Firstly, in Figure 13a, we revisit the examples from Figures 6a and 6b, coloring each sample based on the value of $P(z)$. As depicted in Figure 13a, $P(z)$ proves to be an efficient metric to separate

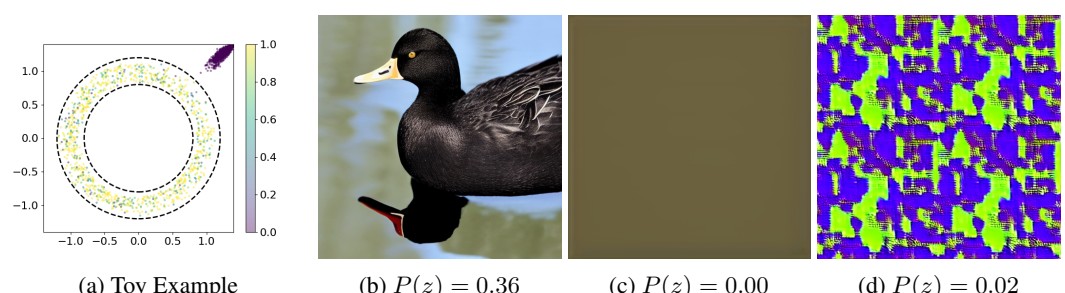

(a) Toy Example    (b) $P(z) = 0.36$    (c) $P(z) = 0.00$    (d) $P(z) = 0.02$

Figure 13: Examples of generated samples with corresponding values of $P(z)$.

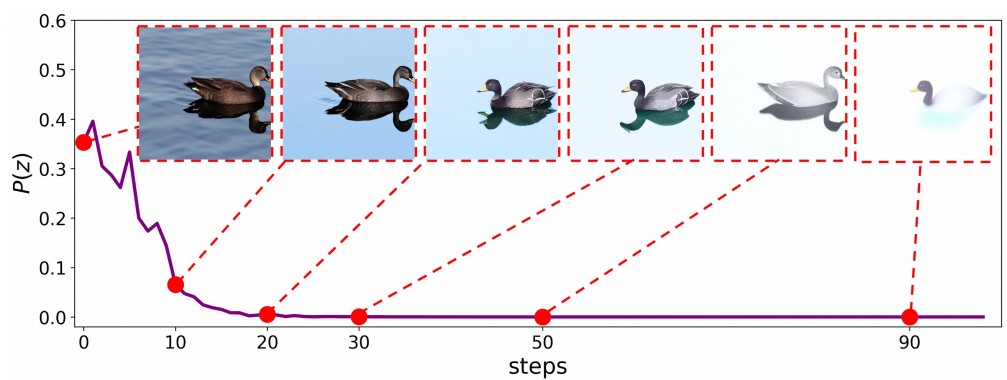

Figure 14: Trajectory of $P(z)$ on optimizing brightness reward.

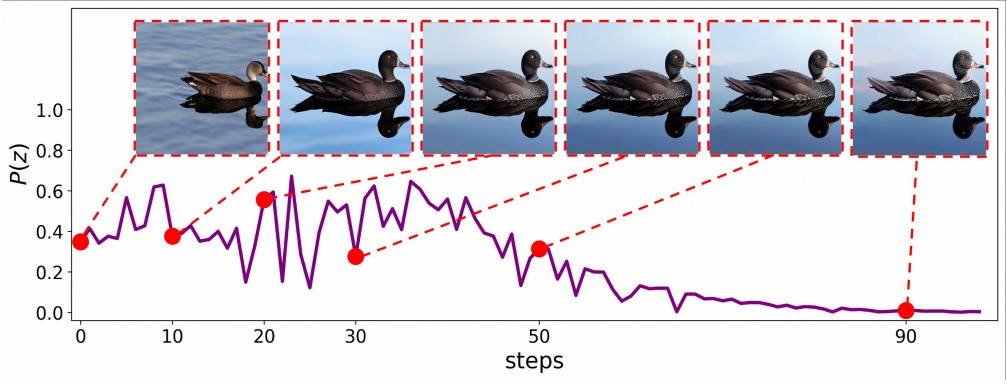

Figure 15: Trajectory of $P(z)$ on optimizing Aesthetic Score.

in-distribution samples from out-of-distribution samples; those in-distribution have high values for $P(z)$, whereas those out-of-distribution exhibit values of $P(z)$ near zero.

Secondly, we manually construct several noise vectors that reside in the low-probability region of the standard Gaussian distribution. To establish a baseline comparison, we first draw one sample from the standard Gaussian distribution and use it to generate an image with Stable Diffusion v1.5 and the prompt "black duck". As can be seen, this leads to a normal image with $P(z)$ also within a reasonably large value. We then construct the low-probability vectors in two ways. The first one is to use all-zero vectors, which obviously reside in the low-probability zone of high-dimensional Gaussian distributions. The generated images with all-zero vectors are visualized in Figure 13c, showcasing that there is nothing discernible in the image while $P(z)$ approximates zero. The second method is to repeat parts of the noise vectors, such that the noise vectors exhibit high covariance in the elements. Specifically, we construct the repeated vectors by first generating an $n/4$ dimensional $z_0$ from the standard Gaussian distribution, and then constructing the noise vectors as $z = [z_0, z_0, z_0, z_0]$, making $z$ an $n$ dimensional vector. The figure corresponding to these repeated vectors, shown in Figure 13d, once again results in a poor image, with $P(z)$ illustrating that the noise vectors also come from a low-probability region.

We further visualize the entire optimization trajectory for the examples in Figures 6c and 6d, i.e., optimizing the brightness reward for Stable Diffusion v1.5 with the prompt "black duck" in Figure 14. Specifically, from Figure 14 we can clearly see that the value of $P(z)$ gradually decreases, and the generated image also gradually diverges from the distribution associated with a black duck. Notably, at around 20 steps, the value of $P(z)$ becomes near-zero, and at the same time, the generated image more closely resembles a blue duck rather than the specified black duck.

Similarly, we visualize the optimization trajectory for optimizing the Aesthetic Score for SD v1.5 with the prompt "black duck". The results are in Figure 15. A clear conclusion is that in this case, it is less likely for the optimized samples to be out-of-distribution. This is mainly because the Aesthetic Score itself penalizes those OOD samples. It is noteworthy to observe that this insight is also captured by our proposed indicator $P(z)$, because when comparing the trend of $P(z)$ in Figure 14 and Figure 15, we can see that optimizing the Aesthetic Score leads to a much less significant decrease in the $P(z)$ value.

## E  IMPLEMENTATION DETAILS

In this section, we discuss some implementation details of our proposed method, as well as clarify some omissions in the experimental section.

### E.1  ALGORITHM IMPLEMENTATION

It is clear that to solve the direct noise optimization problem stated in Problem 3, differentiation of the noise-to-sample mapping $M_\theta$ is required. It is worth noting that this differentiation cannot be handled by standard auto-differentiation in PyTorch (Paszke et al., 2019), as it can lead to a memory explosion. A common technique to resolve this issue is gradient checkpointing, which has also been adopted by other related works on noise optimization (Wallace et al., 2023b; Novack et al., 2024; Karunratanakul et al., 2023).

Here, we describe an efficient method to implement our proposed hybrid gradient estimators detailed in Section 4, along with the optimization process, by utilizing the built-in auto-differentiation in PyTorch (Paszke et al., 2019). Specifically, suppose we wish to use $q$ samples to estimate the gradient in Equation equation 12; that is, we draw $q$ noise vectors for perturbation: $\xi_1, ..., \xi_q$. We then generate the corresponding samples $x_i = M_\theta(z + \mu\xi_i)$ for $i = 1, ..., q$. At this point, we should compute the estimated gradient of the reward functions in a non-differentiable mode as follows:

$$\hat{H}_2(x) = \frac{1}{q}\sum_{i=1}^{q}(r(M_\theta(x_i)) - r(x))(x_i - x). \tag{17}$$

Finally, we can execute gradient backpropagation with the loss function,

$$loss(z) = \langle \hat{H}_2(x), M_\theta(z) \rangle,$$

which produces the exact gradient estimator for $z$.

| Method | LGD | | | BoN | | | DNO |
|---|---|---|---|---|---|---|---|
| | $n$=10 | $n$=100 | $n$=1000 | $n$=10 | $n$=100 | $n$=1000 | 5 min |
| Darkness ↑ | 158 | 165 | 169 | 153 | 155 | 156 | 241 |
| CS ↑ | 26.1 | 26.0 | 26.0 | 26.4 | 26.2 | 26.2 | 25.6 |
| ITM ↑ | 98 | 98 | 97 | 98 | 100 | 96 | 94 |

Table 3: Performance comparison for optimizing darkness. The experimental setting uses the average performance over 100 random prompts and seeds.

| Method | LGD | | | BoN | | | DNO |
|---|---|---|---|---|---|---|---|
| | $n$=10 | $n$=100 | $n$=1000 | $n$=10 | $n$=100 | $n$=1000 | 5 min |
| Brightness↑ | 167 | 173 | 172 | 161 | 169 | 169 | 246 |
| CS ↑ | 25.9 | 26.0 | 25.7 | 26.1 | 26.2 | 26.1 | 25.7 |
| ITM ↑ | 94 | 98 | 96 | 98 | 99 | 96 | 95 |

Table 4: Performance comparison for optimizing brightness. The experimental setting uses the average performance over 100 random prompts and seeds.

### E.2 EXPERIMENT DETAILS

In this section, our goal is to provide the experimental details that were omitted from Sections 5.1, 5.2, 5.3.

**Details for Section 5.1.** In this experiment, to solve the probability-regularized noise optimization problem as formulated in Equation equation 9, we employ the Adam optimizer (Kingma & Ba, 2014) with a learning rate of 0.01. For optimization with regularization, we set the regularization coefficient $\gamma$ to 1. To compute the minibatch stochastic gradient for the regularization term in Equation equation 9, we set the batch size $b$—the number of random permutations drawn at each step—to 100. For each optimization run, we utilize a single A800 GPU, with the total memory consumption being approximately 15 GB.

We provided the performance of LGD and BoN on the brightness and darkness tasks and compare them to our DNO with regularization, as shown in the following Table 3 and Table 4. The experimental setting is similar to that in Section 5.1, but we use the average performance over 100 random prompts and seeds instead of 1000 to enable faster simulation. From the results, it is evident that LGD and BoN fail to optimize brightness and darkness to a high level, while, as expected, they also do not encounter the problem of reward hacking even without any regularization technique.

**Details for Section 5.2.** In this second set of experiments, we continue using the Adam optimizer with a learning rate of 0.01. For optimization with regularization, though, we reduce the regularization coefficient to $\gamma = 0.1$ because optimizing these human-like reward functions is less susceptible to the OOD reward-hacking issue, while maintaining the batch size for the permutation matrix $b$ at 100. Each optimization run also uses a single A800 GPU, but the total memory consumption is around 20 GB. For the experiment in Section 5.2, we use a prompt set similar to those in prior works such as DDPO and AlignProp. The prompts take the form of "<color> <animal>", where "<color>" is randomly selected from a color list and "<animal>" is randomly selected from an animal list. For example, a sample prompt could be "purple duck." Regarding the baselines in Table 1, we implemented LGD (Song et al., 2023a) ourselves, following the algorithm from their paper on these reward functions. In the experimental setting of Table 1, $n = 100$ is a fairly time-consuming setting for LGD, as it requires backpropagation through the neural network 100 times for each diffusion step, which takes approximately 7 minutes to complete. For other baselines, we reuse the statistics presented in their corresponding papers.

To analyze the performance of our DNO algorithm and the best tuning-based algorithm, AlignProp on the performance on the OOD reward-hacking issue. We used test metrics in Figure 3 to measure the level of OOD, similar to what we did for DNO. As shown in the Table 5 and 6, AlignProp does exhibit a certain degree of reward hacking at higher OOD levels, but the effect remains within an acceptable range.

| Method | SD v1.5 | AlignProp | DNO | | |
| --- | --- | --- | --- | --- | --- |
| | | | (1 min) | (3 min) | (5 min) |
| Aesthetic ↑ | 5.367 | 8.940 | 5.754 | 7.202 | 8.587 |
| HPS ↑ | 0.278 | 0.266 | 0.276 | 0.272 | 0.271 |

Table 5: Table for optimizing Aesthetic Score, with HPS Score as test metric

| Method | SD v1.5 | AlignProp | DNO | | |
| --- | --- | --- | --- | --- | --- |
| | | | (1 min) | (3 min) | (5 min) |
| HPS ↑ | 0.278 | 0.330 | 0.285 | 0.303 | 0.324 |
| Aesthetic ↑ | 5.367 | 5.060 | 5.253 | 5.519 | 5.311 |

Table 6: Table for optimizing HPS Score, with Aesthetic Score as test metric

Here we provide additional quantitative experiments using DNO with SD v1.5 to investigate whether DNO performs well with complex prompts. We tested 100 prompts from the Pick-a-Pic test dataset, similar to the setting used in SPIN (Yuan et al., 2024). Table 7 compares the performance of SPIN (quoted from Table 3 in (Yuan et al., 2024)) with the average performance of our DNO. As shown, DNO performs well on complex prompts. This result is not surprising, as DNO is designed to optimize noise vectors specific to each prompt, ensuring robust performance across diverse scenarios.

**Details for Section 5.3.** In this section, the primary hyperparameters for the three tested algorithms are the perturbation coefficient $\mu$ and the number of samples $q$ used to approximate the gradient (as formulated in Equation equation 17). Clearly, $q$ plays a crucial role in determining the running time of each algorithm. For an equitable comparison, we tune $q$ separately for each algorithm to achieve roughly the same time cost per gradient step. Specifically, we set $q$ values for ZO-SGD, Hybrid-1, and Hybrid-2 to 16, 8, and 4 respectively. For $\mu$, we also adjust them individually for each algorithm, as they have varying sensitivity to $\mu$. Through trial and error, we select $\mu$ values of 0.01 for ZO-SGD and Hybrid-1, and 0.02 for Hybrid-2. Finally, for optimizing JPEG Compressibility, we use the Adam optimizer with a learning rate of 0.01, but for the Aesthetic Score experiment, we reduce the learning rate to 0.001, as we found that 0.01 can lead to divergence during optimization for the Aesthetic Score. Each optimization run continues to use a single A800 GPU.

We also provide a discussion on the time cost of non-differentiable methods as follows:

For ZO-SGD, the time per gradient step is roughly the same as generation time, which is 2s, but it does not work effectively in practice. For Hybrid-1 and Hybrid-2, since these methods also require differentiating through , the time per gradient step is comparable to DNO in the differentiable setting from Section 5.2—approximately 7 seconds per gradient step. From Figure 4, we observe the following: For Hybrid-2, it can optimize JPEG Compressibility to a reasonably high level (-30 to -20) within 5 minutes. For Aesthetic Score, it takes around 20 minutes to reach a reasonably high score (>7), which is significantly slower than using gradient information. However, this experiment was included primarily as a showcase to demonstrate that Hybrid-2 can still optimize effectively without gradient information.

## F HYPERPARAMETERS ANALYSIS

In this section, we conduct a thorough analysis of the hyperparameters for the proposed method. Our objective is to offer a concise guideline for selecting the hyperparameters in the proposed method.

| Method | SPIN | DNO | | |
| --- | --- | --- | --- | --- |
| | | (1 min) | (3 min) | (5 min) |
| Aesthetic ↑ | 6.248 | 6.013 | 6.993 | 8.305 |
| HPS ↑ | 0.276 | 0.279 | 0.291 | 0.326 |
| PickScore ↑ | 22.00 | 21.85 | 23.61 | 24.89 |

Table 7: Running DNO with SD v1.5 and Prompts from Pick-a-Pic Test dataset

As discussed in Section 3.1, the concentration inequalities involve a hyperparameter $k$, which represents the dimension of subvectors from the noise vectors $z$ that we aim to assess probabilistically. As noted in Remark 2, the dimension $k$ should be neither too large nor too small. Additionally, another critical hyperparameter is the number of permutation matrices $b$ employed to compute the stochastic gradient for the probability regularization in Equation equation 9. Furthermore, we aim to explore the impact of the regularization coefficient $\gamma$ in the probability regularization term.

To examine the effects of $k$, $b$, and $\gamma$ on mitigating the OOD (Out-Of-Distribution) reward-hacking problem, we revisit the experiment of optimizing darkness reward with the prompt "white <animals>" from Section 5.1. In Figure 16, we illustrate how these three hyperparameters influence both the reward and the consistency score (CS), across four different values.

Firstly, Figure 16a supports the notion that $k$ should be carefully chosen—not too large, yet not overly small. We observe that $k = 1$ underperforms compared to $k = 2$ and $k = 10$, as selecting $k = 1$ fails to account for the covariance among noise vectors. Conversely, $k = 100$ proves to be a poor choice because it entails a smaller $m$, potentially rendering the concentration inequalities detailed in Lemma 1 less precise.

Secondly, as demonstrated in Figure 16b, the number of permutation matrices $b$ seems to have a minor impact on the optimization process, provided $b$ is sufficiently large. Based on empirical evidence, $b = 100$ emerges as an optimal selection for the proposed method.

Lastly, the effects of $\gamma$ are depicted in Figure 16c. Adjusting the value of $\gamma$ clearly presents a trade-off between convergence speed and the propensity for OOD reward-hacking problems. Given this observation, we recommend empirically tuning the value of $\gamma$ for different reward functions and prompts using a limited number of samples and a few optimization steps.

## G  LITERATURE REVIEWS

First of all, we will briefly review several recent and important references that are related to this work. (Uehara et al., 2024a) is a survey paper which provide a more unified framework for the RL-based methods (Black et al., 2023) and (Fan et al., 2023). (Barceló et al., 2024) investigates the mode collapse problem, which is another important challenge for aligning the diffusion model aside of the OOD reward-hacking problem studied in this work. (Rector-Brooks et al., 2024) is a concurrent work that inference-time optimization for discrete diffusion models, different from the continuous diffusion models that are explored in this work. (Li et al., 2024) is another concurrent work, which addresses inference-time optimization for both discrete and continuous diffusion models. (Li et al., 2024) is conceptually similar to (Song et al., 2023a) and includes comparisons with BoN and DPS. AlignProp (Prabhudesai et al., 2023) and DRAFT (Clark et al., 2023) are two concurrent works proposing essentially the same algorithm, that is directly fine-tuning the diffusion models with the gradient of differentiable reward functions.

## H  SMOOTHNESS JUSTIFICATION FOR PIXEL-BASED REWARD FUNCTION

Conceptually, it is quite straightforward to argue that the reward function is smooth with respect to pixel changes. Small changes to the image pixels would not result in large differences in the reward function's score. In practice, most reward functions exhibit this property, including metrics such as darkness, brightness, compressibility, and even human-preference-based reward functions. Minor pixel changes generally do not cause significant differences in the reward evaluation process.

To provide a more concrete answer, we conducted a quantitative analysis. Here is the setup we adopted: We first sampled a noise vector $x_1 \sim \mathcal{N}(0, I)$ and then generated a second noise vector $x_2$ in the neighborhood of $x_1$, i.e., $x_2 \sim \mathcal{N}(\sqrt{0.9}x_1, \sqrt{0.1}I)$. Using the Aesthetic Score as the reward function $r(\cdot)$, we computed the following quantities:

$$A = E_{x_1,x_2} \frac{|r(M_\theta(x_1)) - r(M_\theta(x_2))|}{||x_1 - x_2||}$$

and

$$B = E_{x_1,x_2} \frac{||\nabla r(M_\theta(x_1)) - \nabla r(M_\theta(x_2))||}{||x_1 - x_2||}.$$

Using 100 samples with random prompts from Section 5.1, we estimated these values to be $A = 0.19$ and $B = 6.93$, respectively. These results rigorously demonstrate that the composite mapping $r \circ M_\theta$ is indeed smooth. We will include this justification in a more formal way in the revised manuscript.

It is worth noting that for any function $f$, as long as its gradient norm is bounded by a constant $L/2$, the function is $L$-smooth. In practice, assuming a bounded gradient norm is not restrictive for reward functions, especially those based on neural networks, given the presence of many normalization layers.

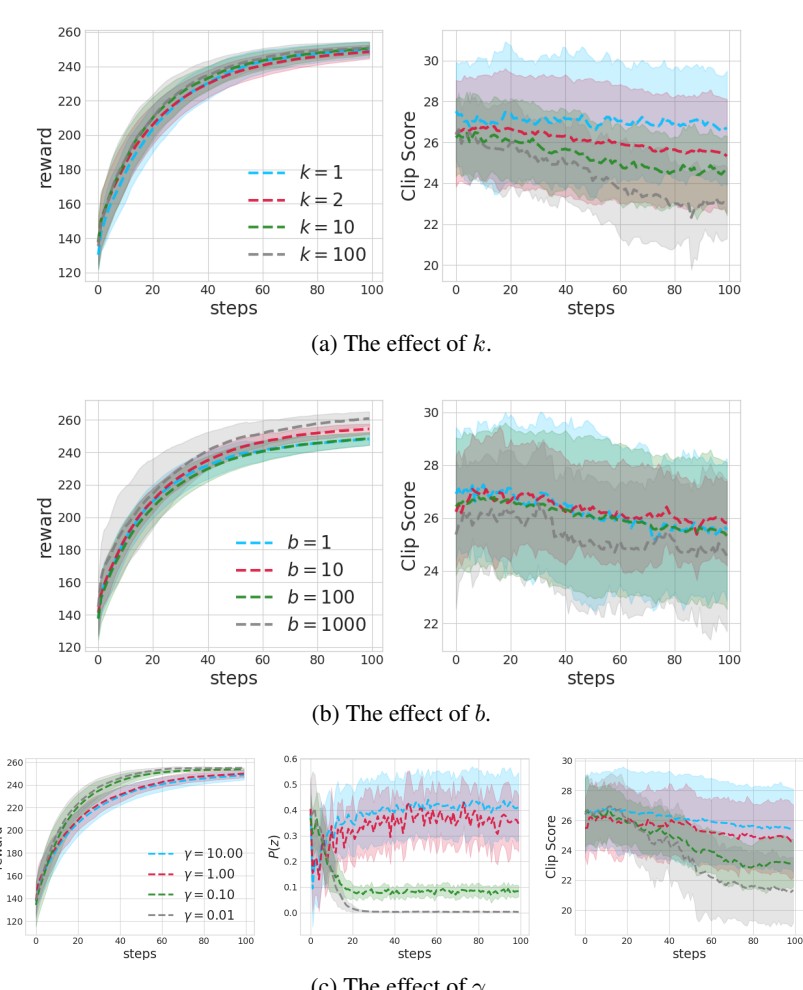

(a) The effect of $k$.

(b) The effect of $b$.

(c) The effect of $\gamma$.

Figure 16: Hyperparamet Analysis

