# OpenReview forum: "Inference-Time Alignment of Diffusion Models with Direct Noise Optimization"
_ICLR.cc/2025/Conference — Submitted to ICLR 2025_

### Official Review · Reviewer_7yop · 2024-11-02

**Soundness:** 3
**Presentation:** 3
**Contribution:** 3
**Rating:** 6
**Confidence:** 4

**Summary:**

The paper addresses the alignment of diffusion models to downstream tasks with continuous reward functions. It proposes Direct Noise Optimization (DNO) as a tuning-free, inference-time method for adjusting generated samples to maximize target rewards. Moreover, in the face of reward hacking (aka over-optimization), the paper introduces a probability regularization for alleviation.

**Strengths:**

1. DNO's main feature is training-free. Current researchers have been more interested in inference-time techniques that do not require costly fine-tuning. Therefore, this work stands for an important and attentive direction of aligning generative models.
2. The integration of probability regularization to mitigate out-of-distribution reward hacking is interesting.
3. The authors applied DNO across various reward functions, highlighting its adaptability.
4. DNO is compared rather comprehensively with some baselines on a variety of reward functions, which is good.

**Weaknesses:**

1. While theoretical parts of the probability regularization are comprehensive, I find those parts hard to parse. Could the authors highlight the key message of such regularization, and why would it work from a high-level perspective?
2. Reward hacking is known to be a key concern in aligning LLM/diffusion models. The authors attempt to alleviate hacking through a regularize, which is good. However, the evaluation way remains problematic. In Fig 2, the authors presented that adding regularize sacrifices little performance for retraining fidelity. However the authors did not compare regularized DPO with other methods. In the end, the observation of Fig 2 is not a surprise to me because adding regularization almost surely leads to a tradeoff of performance and fidelity. There exist some works showing that simply adding small regularization to align-prop would greatly alleviate reward hacking. Therefore, Fig. 2 does not provide any further impressive information.
3. One primary concern is that this work lacks many comparisons with existing fine-tuning-based and inference-based techniques that appear earlier or concurrent to this work. See more details below.

**Questions:**

1. How does DNO perform when used with other diffusion sampling methods besides DDIM? From my perspective of view, I think DDIM is the easiest and most direct way to model the mapping from noise to sample, i.e. $M_\theta(z)$. If using other schedulers other than DDIM, I wonder how DNP will work to simulate $M_\theta$. Moreover, if the algorithm leverages a differentiable reward function for optimization, the gradients are passed backward through all diffusion steps. I assume that would bring significant computational costs comparable to direct reward propagation. Can the authors comment on this?

2. I don't see how eq (2) brings a "prompt-agnostic" method. For example, when aligning stable diffusion, is the optimized initial noise a "universal" optimal value for all prompts?

3. Currently, there are already many works aimed at aligning diffusion models. Roughly, they can be classified into two categories,
      (1) directly fine-tune the diffusion models. See:  https://arxiv.org/abs/2407.13734, https://arxiv.org/abs/2410.08315.   While DNO includes comparisons with alignprop and ddpo in Table 1, comparisons with many later RL-based finetuning methods that alleviate reward hacking issues are missing. I wonder how DNO would work if compared to them.
      (2) purely inference time techniques. While DNO does not fine-tune diffusion models, DNO still needs to optimize through the diffusion pipeline to get optimized noise. The computational costs are still heavy. Here are some works that are purely inference-time and prompt-agnostic: https://arxiv.org/abs/2410.08134, https://arxiv.org/abs/2408.08252.

Discussing and comparing these works quantitatively (or qualitatively) is important for positioning DNO. However, it might also be true that these works are concurrent with this submission. Finally, at least qualitative discussions are needed.

4. I assume the numbers in Table 1 do not consider reward hacking which is problematic for a method that claims to alleviate reward hacking. This is because alignprop would significantly suffer from mode collapse when aesthetic score > 8.5 from my experience. But Table 1 reports 8.9 for alignprop. Can the authors comment on this?

---

> ### Comment · Reviewer_7yop · 2024-11-23
> **Thanks for the informative responses**
>
> Upon clarifications, my concerns are mostly addressed. The authors' responses have (1) clearly indicated the generalization capacity of DNO especially on its prompt-agnostic feature (in the original version, it brought much confusion); (2) explained the evaluation pipeline conducted in this work in the face of reward hacking and (3) positioned DNO with the diffusion model alignment literature. I think I will keep my current score. For clarity of presentation, I recommend the authors incorporate such important clarifications, especially regarding the algorithmic/implementation details and discussions of previous works in the next version.

---

### Official Review · Reviewer_c2Cr · 2024-11-04

**Soundness:** 3
**Presentation:** 2
**Contribution:** 2
**Rating:** 6
**Confidence:** 2

**Summary:**

This paper proposes Direct Noise Optimization (DNO), a novel approach for aligning diffusion models with continuous reward functions at inference time. The key idea is to optimize the injected noise during the sampling process to maximize desired reward functions, without needing to fine-tune the model parameters. The authors develop theoretical foundations for DNO, introduce probability regularization to prevent out-of-distribution samples, and extend the method to handle non-differentiable rewards. Extensive experiments demonstrate that DNO can achieve state-of-the-art reward scores within reasonable inference time budgets.

**Strengths:**

1. The paper provides a thorough treatment of DNO, including theoretical analysis, practical implementation details, and solutions to key challenges (OOD reward hacking, non-differentiable rewards). The probability regularization technique using concentration inequalities is particularly novel and well-motivated since it addresses important practical concerns without no fine-tuning, works with arbitrary reward functions, can handle non-differentiable rewards, etc.
2. The experimental evaluation is extensive and convincing, showing that DNO can match or exceed the performance of fine-tuning methods while requiring only inference-time optimization. The authors test on multiple reward functions and provide detailed ablation studies.

**Weaknesses:**

1. Although the paper proposes a regularization technique to provide overfit, the resulting images still seem to overfit. Besides, the regularization only focuses on constraining the initial noise distribution to the high probability area. Also see question 1
2. Lack of baseline results: There is no inference time cost comparison with LGD since it's an important inference-time method as the baseline. Besides, a more relevant training method is DRAFT [1], which also uses direct backpropagation to optimize the reward function. It also faces a similar issue of reward hacking as the proposed method.


[1] Directly Fine-Tuning Diffusion Models on Differentiable Rewards
Kevin Clark, Paul Vicol, Kevin Swersky, David J Fleet

**Questions:**

1. In fact, the model is repeatedly applied to intermediate noises too. Why there is no regularization of out-of-distribution intermediate noise inputs to the model?
2. Why there is no comparison to LGD and other baseline in Section 5.1? The performance of LGD is surprisingly bad in Section 5.2, even worse than SD 1.5. The authors argue that the reason is the complex reward used in 5.2, and I wonder whether LGD has a better performance in similar reward functions like 5.1.

======================

Post rebuttal:
The author rebuttal solved my concern about the motivation of the regularization technique. However, I feel like the paper still needs lots of improvement on writing and more comprehensive experiments. So I will raise my score to 6 but with lower confidence.

---

### Official Review · Reviewer_ZzXe · 2024-11-05

**Soundness:** 2
**Presentation:** 3
**Contribution:** 2
**Rating:** 5
**Confidence:** 3

**Summary:**

This paper focuses on the alignment of diffusion models by optimizing noises in the sampling process. The authors propose to optimize noises at each sampling step to improve the reward value, rather than optimize the model parameters. To this end, they discuss the effectiveness of the proposed DNO method, and address the reward hacking problem in optimization via a regularization term. Experiments show that DNO improves the reward value of generations.

**Strengths:**

+ This study considers aligning diffusion models in the inference time, alleviating the computational cost of aligning the model.
+ The authors identify and address the problem with reward hacking in optimization.
+ The proposed method is applicable for both differentiable and non-differentiable reward functions.

**Weaknesses:**

- It seems that the idea of optimizing noises has been proposed by (Wallace et al., 2023b; Ben-Hamu et al., 2024; Novack et al., 2024; Karunratanakul et al., 2023). What is the essential difference between DNO and these prior works, except that DNO optimizes noises at each step?
- The assumption of Theorem 1 is not convincing. I am not sure how to understand the smoothness from Figure 4 (Tang et al., 2024a). Although the noise-to-sample mapping is smooth, the reward function is usually complicated, especially for reward functions related to human preference. Is there any direct justification for the smoothness of $r\odot M$?
- The faster convergence speed of DNO is not well supported by their theoretical analysis. Equation (5) only demonstrates the final performance of DNO with SDE may be better than DNO with ODE, but does not justify the optimization speed.
- In equation (10), there should be a $1/\mu$ term.
- There are several concerns about the experiment in Table 1. First, it is unclear what prompts and how many generations are used for evaluation. Second, for DNO, the annotated time is confusing. Does ``1 min’’ mean that generating one image using DNO costs 1 minute? Third, how about the performance of DNO with ZO-SGD, hybrid-1, and hybrid-2?
- The evaluation for reward hacking based on $P(z)$ is limited. $P(z)$ only indicates the distribution of initial noise $z$ but cannot describe the distribution of the final generated image. I’d like to see more comparisons between images generated by different methods, either quantitatively or qualitatively.
- Which experiment supports the claim in Line 452 `` while it prevents the test metrics, i.e., the HPS score and Pick Score, from decreasing throughout the process'’?
- Is the proposed DNO also effective for more complex prompts? Have the authors compared images generated by different methods on both simple and complex text prompts?

**Questions:**

Please refer to Weakness part.

---

### Official Review · Reviewer_DJei · 2024-11-10

**Soundness:** 3
**Presentation:** 3
**Contribution:** 2
**Rating:** 5
**Confidence:** 2

**Summary:**

This paper introduces a method called direct noise optimization (DNO) for maximizing some reward function for the generated samples. Different from finetuning and RL methods, DNO operates as a form of testing-time optimization. DNO is also extended to the non-differentiable reward setting by leveraging zero-order optimization techniques. Some theoretical analyses are provided to account for its empirical effectiveness.

**Strengths:**

- The idea is simple and neat. Directly optimizing the noise towards samples of maximum reward value is technically sound. The paper presents the idea in a clear way.

- The paper considers both differentiable and non-differentiable reward settings, making direct noise optimization a quite practical method to use.

- The paper provides some interesting theoretical insights to justify the drawback of direct noise optimization. Moreover, it provides a solution to avoid this drawback with a probability regularization.

**Weaknesses:**

- As the paper mentioned, directly optimizing the input noise can result in degenerated solutions, which the authors describe as OOD reward hacking. I think the problem, despite being alleviated by some additional regularization, will fundamentally limit the generation quality. This is conceptually similar to the problem of adversarial examples. Moreover, regularizing the noise to lie in a high-probability region may alleviate the reward hacking problem, but it also limits the ability to maximize the reward function. How can the effectiveness of this probility regularization be measured in practice? How to set the hyperparameter $\gamma$ in practice?

- Zero-order optimization is applied to the non-differentiable reward setting, but I am highly skeptical about its empirical effectiveness, as the zero-order optimization is known to be difficult to converge, not to mention that the noise optimization could be very challenging and non-convex/non-smooth.

- The performance gain of DNO over the other baselines also seems be quite marginal from Table 1. For the other baseline methods, there are many hyperparameters that can be tuned. I think to better illustrate the performance gain, it will be good to compare the performance with different finetuning steps (or different inference-optimization steps).

**Questions:**

See the weakness section.

---

### Meta-Review · Area_Chair_9EcD · 2024-12-21

**Metareview:**

This paper presents a method called *Direct Noise Optimization* (DNO) for maximizing a reward function associated with generated samples. Unlike finetuning or reinforcement learning approaches, DNO functions as a test-time optimization technique. The method is further extended to handle non-differentiable reward settings using zero-order optimization techniques. By addressing both differentiable and non-differentiable reward scenarios, DNO demonstrates practical applicability.

The concept of optimizing noise has been previously explored in works by Wallace et al. (2023b), Ben-Hamu et al. (2024), Novack et al. (2024), and Karunratanakul et al. (2023). Additionally, related ideas, such as using diffusion models for black-box optimization (ICML 2023, NeurIPS 2024) and multi-objective optimization (ECML 2024), exhibit similarities to this approach. Consequently, the technical novelty of this work appears limited.

Reviewers have raised concerns about the rationale behind regularizing noise vectors alone. For instance, maintaining all intermediate additive noise distributions while varying the initial noise distribution leaves the effect on the final output distribution unclear. Another critical issue is the lack of comparisons with prior fine-tuning-based and inference-based methods, whether preceding or concurrent to this work.

The paper would benefit from significant improvements in writing clarity and more comprehensive experimental evaluations to establish its contributions more robustly.

**Additional Comments On Reviewer Discussion:**

There was an extensive discussion between the reviewers and the authors, particularly regarding the core idea of regularizing noise vectors. One key concern remains unresolved: why does regularizing the noise vectors alone make sense? For instance, if all intermediate additive noise distributions are kept constant, it is unclear how changes in the initial noise distribution would impact the final output distribution. Would the impact be bounded solely by the changes in the initial noise distribution? Without considering the actual distribution of the intermediate outputs, there appears to be no established upper bound to confirm this.

The authors were unable to fully address these concerns. Additionally, the lack of comparisons with existing fine-tuning-based and inference-based methods, whether preceding or concurrent, was another significant issue. As noted by the reviewers, the paper requires substantial improvements in both writing clarity and the inclusion of more comprehensive experiments to strengthen its contribution.

---

### Decision · Program_Chairs · 2025-01-22

Reject